# Examination Feedback Simulation: Beyond Static and Unique Clinical Trajectories

## Abstract

Large language models (LLMs) have shown significant promise in healthcare applications. To better mirror real-world settings for LLM evaluation, dynamic longitudinal diagnosis-and-treatment simulation with virtual patients has recently emerged as a focal point of research. However, existing simulation frameworks are constrained by the limitation of clinical trajectory uniqueness, where virtual patients can only provide feedback based on information available in static electronic health records (EHRs). This limitation leads to simulation failures when unrecorded but medically sound examinations are ordered. In this paper, we formulate the task of Examination Feedback Simulation to address this limitation, which aims to dynamically augment the unique trajectory by simulating medically plausible examination results in response to clinical orders. To support this largely unexplored research, we construct ClinTrack, a dataset curated from MIMIC-IV. ClinTrack is organized in a hierarchical, chronologically-ordered structure to facilitate sequential clinical tasks. We further propose a structure-aware evaluation metric SimScore to quantitatively assess the quality of simulated results, which shows promising initial alignment with expert judgment. Building on this framework, we develop ClinSim, a new generative model specifically designed for this task. Experiments demonstrate that our 4-billion-parameter ClinSim model significantly outperforms flagship models up to 235B parameters on this task, achieving an improvement of over 10 percentage points in SimScore, providing a critical foundation for creating more dynamic and realistic virtual patients.

## 1 Introduction

Artificial intelligence (AI), particularly large language models (LLMs), is considered a promising solution to address the long-standing challenges in healthcare, such as access, quality, and cost containment (Kissick, 1994; Wang et al., 2024b; Achiam et al., 2023; Brodeur et al., 2024; Singhal et al., 2023). To provide a more realistic basis for evaluating LLM performance, dynamic longitudinal diagnosis-and-treatment simulation has recently emerged as a focal point of research, as it is a crucial step towards mirroring real-world clinical practice (Nori et al., 2025; Zhou et al., 2025; Shang et al., 2025; Li et al., 2024; Fansi Tchango et al., 2022; Dou et al., 2025; Liu et al., 2025b; Johri et al., 2024; Liu et al., 2025a). In these simulation frameworks, the medical LLM agent acts as a doctor, and another LLM agent serves as a virtual patient that simulates the actions of sequentially reporting symptoms and examination feedback to the doctor by referring to electronic health records (EHRs). After the simulation, the performance of the medical LLM is evaluated based on the entire clinical interaction process with the virtual patient. Among various clinical actions, examinations (including radiological, microbiological, and laboratory examinations) are one of the most essential, as the ability to order and interpret them is a fundamental component of medical competence (Zayed et al., 2025; Amrollahi et al., 2025; Johnson et al., 2019).

However, the simulation of medical examinations in prior studies is inadequate. Existing simulation frameworks inherit the limitation of clinical trajectory uniqueness naturally present in static EHRs, implicitly excluding any potentially reasonable deviation from the recorded clinical actions. This limitation, illustrated in Figure 1, creates a critical failure point: as existing virtual patients can only provide examination feedback based on the unique and static EHRs (Zhou et al., 2025; Liu et al., 2025b), ordering any unrecorded examination, even if medically sound, can result in the interaction's

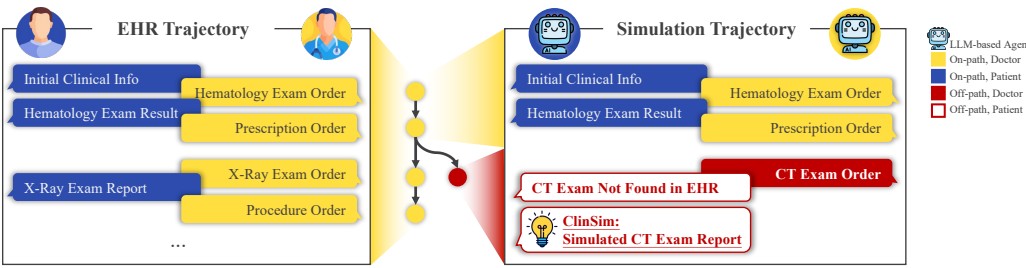

Figure 1: The limitation of clinical trajectory uniqueness in static EHRs. By referring to EHRs (left), the virtual patient in simulation frameworks (right) provides feedback to the LLM doctor. It functions well (top right) until the doctor orders an unrecorded examination (bottom right). Our approach addresses this limitation by enabling the virtual patient to simulate medically plausible examination results for any reasonable examination orders.

inability to proceed (Nori et al., 2025; Shang et al., 2025). The virtual patient's refusal to provide feedback on unrecorded examinations degrades the evaluation from a genuine assessment of clinical competence to a mere test of whether the LLM doctor can guess the specific examinations listed in the original record.

To address this limitation, we present a comprehensive, three-part solution. (1) We formulate the task of **Examination Feedback Simulation**, which involves leveraging a patient's clinical history to simulate medically plausible examination results in response to medical orders. To support this, we introduce ClinTrack-MIMIC (**ClinTrack** for simplicity), a dataset curated from MIMIC-IV (Johnson et al., 2020) for sequential clinical tasks. It is created by converting the source tabular data into a hierarchical, chronologically-ordered structure through a process of data aggregation followed by two-step fine-grained parsing. (2) We propose **SimScore**, an objective, rule-based evaluation metric that leverages the structured nature of our data to quantitatively assess the simulated results. A blind review by a medical expert on a randomly selected sample set confirmed a substantial alignment with SimScore's rankings, underscoring its potential as an objective metric for quantitative evaluation. (3) We develop **ClinSim**, a specialized generative model, to demonstrate the potential of a purpose-built LLM in this task. ClinSim outperforms various few-shot learning flagship models ranging from 32B to 235B by over 10 percentage points on SimScore with only 4 billion parameters.

Our key contributions can be summarized as follows:

- We make the first attempt to systematically define the task of examination feedback simulation, and introduce the supporting ClinTrack dataset curated from MIMIC-IV, laying the groundwork for creating more realistic simulation frameworks.

- We propose SimScore, a rule-based metric derived from a structure-aware evaluation schema, to assess the quality of the simulated results, and investigate its alignment with expert judgment in an initial human evaluation.

- We develop a new generative model ClinSim specifically designed for this task with superior performance compared to existing models, demonstrating the potential of LLMs in this task.

## 2 RELATED WORK

**Medical Question-Answering** has been a prominent approach to evaluate medical language models, with numerous datasets developed. Datasets such as MedQA (Jin et al., 2021), MedMCQA (Pal et al., 2022), PubMedQA (Jin et al., 2019), MedBullets (Chen et al., 2025), MLEC (Li et al., 2021), and MedExpQA (Alonso et al., 2024) primarily assess general medical knowledge, whereas others like CMB-Clin (Wang et al., 2024b), MedXpertQA (Zuo et al., 2025), and EHRNoteQA (Kweon et al., 2024) provide detailed clinical notes or patient records, challenging models to extract relevant information from complex documents.

**Dynamic Longitudinal Diagnosis-and-Treatment Simulation with Virtual Patients** has gained increasing attention recently for its potential to create more realistic clinical interaction scenarios.

CRAFT-MD (Johri et al., 2024), MMD-Eval (Liu et al., 2025a), PPME-LLM (Sun et al., 2025b), Patient Simulator (Liu et al., 2025b), DDXPlus (Fansi Tchango et al., 2022), MediQ (Li et al., 2024) and Baichuan-M2 (Dou et al., 2025) simulate multi-turn clinical dialogues for information gathering, where the model incrementally collects information from a virtual patient role-played by another model. These approaches reflect the communication challenges encountered in real-world clinical practice. A medical model's diagnostic performance declines significantly when it is required to gather information progressively through a multi-turn dialogue, as opposed to receiving the entire record at once to make a final diagnosis (Johri et al., 2024; Liu et al., 2025a). SD-Bench (Nori et al., 2025), DynamiCare (Shang et al., 2025), and MultiCogEval (Zhou et al., 2025) propose frameworks that allow the LLM doctor to order clinical examinations and treatments, receiving feedback based on real patient records.

**Examination Feedback Simulation** is a latent component of these simulation frameworks. It aims to address the limitation of clinical trajectory uniqueness in static EHRs. Due to the limitation, any attempt to order a reasonable but unrecorded examination inevitably halts the interaction. Early research intentionally sidestepped this challenge by relaxing the requirement of "longitudinal", excluding new examination orders from the simulation scope (Fansi Tchango et al., 2022; Liu et al., 2025b). In later studies, MediQ (Li et al., 2024) and DynamiCare (Shang et al., 2025) instruct the virtual patient to faithfully respond based on the original records, leading to a possible refusal to provide corresponding feedback. Nori et al. (2025) highlight a key weakness of this approach: refusals can imply the ground-truth trajectory and discourage valid alternative ones. To overcome this, they employ a language model to simulate feedback for unrecorded examinations. The simulation is treated as an auxiliary function rather than a primary research focus, and is not systematically and quantitatively validated.

## 3 APPROACH

We present a comprehensive framework for developing and evaluating an examination feedback simulation model. We first curate a dataset named ClinTrack from the EHRs, then propose SimScore based on the intrinsic structure of the data to evaluate the quality of the simulated results, and finally define the training procedure, including supervised fine-tuning and reinforcement learning, to optimize the model for this task.

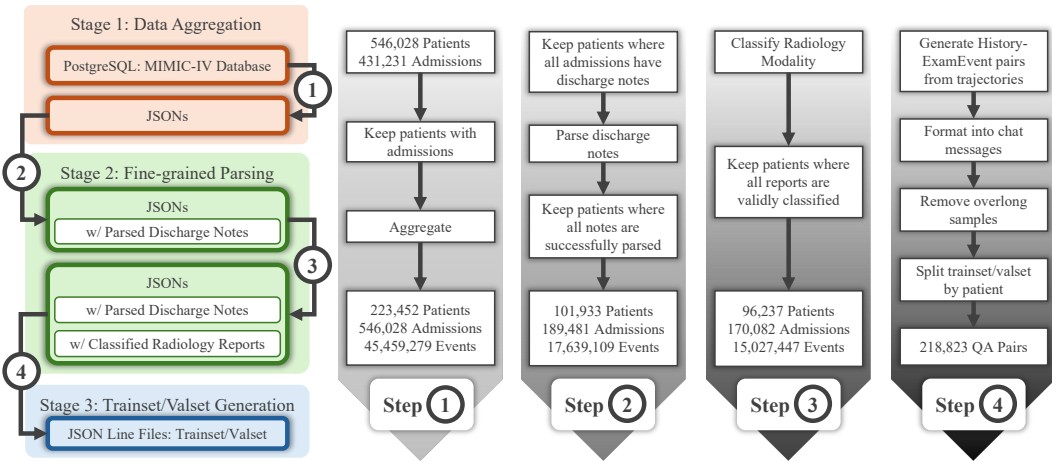

Figure 2: Dataset Curation Process with 3 stages: Data Aggregation, Fine-grained Parsing, and Trainset/Valset Generation. The process is implemented in 4 steps. The JSONs from the first three steps are designed to be upwardly compatible, allowing the records from Step 2 and Step 3 to be merged back for augmentation. This process can create a maximized dataset of 223,452 patients, where a subset of records is enriched with detailed annotations. For this work, to ensure data integrity, we proceed to Step 4 using only the highest-quality output from Step 3. A detailed data structure description is provided in Appendix D.

## 3.1 DATASET CURATION

Our study is based on MIMIC-IV Johnson et al. (2020), a large-scale, publicly available dataset well-suited for constructing a dynamic longitudinal diagnosis-and-treatment simulation framework. Containing over 546K de-identified hospital admission records for 364K patients, MIMIC-IV is widely used for clinical research and is organized into multiple tables linked by unique patient and admission identifiers. The initial stage of data curation involves gathering information across all tables and reconstructing each patient's clinical history into a chronologically ordered sequence of events. The second stage focuses on fine-grained parsing of the clinical notes to extract relevant information. The third stage involves transforming the structured data into prompts for training and evaluation purposes.

**Step ①** Each record, denoted as $R$, is structured as a tuple containing basic patient information $I_{\text{basic}}$, an unstructured clinical note $I_{\text{note}}$, and a sequence of clinical events $I_{\text{seq}} = \{E_i\}_{i=1}^N$, where $E_i$ represents the $i$-th clinical event in chronological order and $N$ is the total number of events.

$$R = (I_{\text{basic}}, I_{\text{note}}, I_{\text{seq}}) \tag{1}$$

**Step ② ③** To leverage the rich information contained in the unstructured notes, we employ an LLM to extract structured information. For discharge notes, this includes admission information $I_{\text{adm}}$, progress information $I_{\text{prog}}$, and discharge information $I_{\text{disch}}$. Detailed contents of these fields can be found in Appendix C. For radiology reports, we extract modality information $I_{\text{modality}}$.

$$(I_{\text{adm}}, I_{\text{prog}}, I_{\text{disch}}) = \text{LLM}(p_{\text{note\_extract}}, I_{\text{note}}) \tag{2}$$

$$I_{\text{modality}} = \text{LLM}(p_{\text{report\_extract}}, I_{\text{report}}) \tag{3}$$

where $I_{\text{report}}$ is a field in $E_i$ if the event type is *RadiologyEvent*, and $p$ denotes the prompt used for the LLM.

**Step ④** We frame the core functionality of the expected model as a conditional generation task. Given a specific time step $i$, the preliminary information of the patient's record is denoted as $S_{\text{pre}}^{(i)}$.

The model is tasked with generating the subsequent clinical event, $E_i$, conditioned on the preliminary information, and the type and time of that event, where the event type is constrained to one of {*RadiologyEvent*, *MicrobiologyEvent*, *LabEvent*}. Note that each event type is further divided into subtypes, such as *CT*, *X-Ray*, etc. for radiology events, *BloodCulture*, *UrineCulture*, etc. for microbiology events, and *Hematology*, *BloodGas*, etc. for lab events. Subtypes are also provided as part of the input to the model. The task is formulated as follows:

$$\hat{E}_i = \text{LLM}(p_{\text{sim}}, S_{\text{pre}}^{(i)}, \text{Type}(E_i), \text{Time}(E_i)) \tag{4}$$

$$S_{\text{pre}}^{(i)} = (I_{\text{basic}}, I_{\text{adm}}, E_1, E_2, \ldots, E_{i-1}) \tag{5}$$

where $p_{\text{sim}}$ is the prompt used to instruct the model to perform the simulation task.

## 3.2 EVALUATION

Existing evaluation metrics for text generation, such as ROUGE (Lin, 2004) and Levenshtein similarity (Levenshtein, 1966), primarily focus on textual similarity. However, since the output of examination feedback simulation is structured data, traditional metrics are suboptimal for two main reasons: (1) they treat the output as plain text, which overlooks the intrinsic structure and semantics of the JSON data; (2) they fail to treat different fields with varying importance appropriately.

To assess the quality of the output, we propose a comprehensive evaluation framework focused on quantifying the fidelity of the simulated clinical events. This framework is predicated on the hypothesis that held-out factual data can serve as a proxy for evaluating the model's ability to generalize to counterfactual queries, as both are unseen during training and are thus equally challenging. Our evaluation framework centers on the structural similarity function $\sigma(\cdot, \cdot)$. It serves as the evaluation metric to compute the SimScore $s$ and as the reward function to compute the reward $r$ in our reinforcement learning methodology. The function is defined as follows:

$$s = \sigma(J_g, J_t) = \begin{cases} \sigma_{\text{dict}}(J_g, J_t) & \text{if } J_g, J_t \in \text{dict} \\ \sigma_{\text{list}}(J_g, J_t) & \text{if } J_g, J_t \in \text{list} \\ \sigma_{\text{str}}(J_g, J_t) & \text{if } J_g, J_t \in \text{str} \\ \sigma_{\text{num}}(J_g, J_t) & \text{if } J_g, J_t \in \text{num} \end{cases} \tag{6}$$

$J_g$ and $J_t$ are the JSON objects parsed from the ground-truth event $E_i$ and the simulated event $\hat{E}_i$, respectively. $\sigma(\cdot, \cdot)$ recursively computes similarity scores based on the objects' structure and content, adapting its behavior for dictionaries, lists, and primitive types.

**Complexity Weighting** aims to ensure that more complex parts of the JSON structure contribute more to the final score, denoted as $\omega(v)$, where $v$ is any value in the JSON structure.

$$\omega(v) = \begin{cases} \sum_{v_i \in v} \omega(v_i) & \text{if } v \in (\text{dict} \cup \text{list}) \text{ and } v \neq \emptyset \\ 1 & \text{otherwise} \end{cases} \tag{7}$$

The complexity function $\omega(v)$ is designed to address a challenge inherent in our recursive similarity function: the potential for normalization at each level to dilute the significance of deeply nested information. It also ensures greater penalties for missing complex structures compared to simpler ones.

**Dictionary Similarity** is a weighted average of the similarity of their keys and values. Let $\mathbb{K}_g$ and $\mathbb{K}_t$ be the key sets of $J_g$ and $J_t$, respectively. The set of matched key pairs, $\mathbb{M} = (\mathbb{K}_g \cap \mathbb{K}_t) \cup \mathbb{M}_f$, is determined by first finding all exact matches, followed by a fuzzy matching step for the remaining keys based on Levenshtein similarity. The fuzzy matching is performed using a greedy algorithm to ensure one-to-one matching. The threshold for fuzzy matching is set to 0.8 in practice to avoid incorrect matches. The dictionary similarity is defined as:

$$\sigma_{\text{dict}}(J_g, J_t) = \sigma_{\text{key}}(J_g, J_t) \times \sigma_{\text{value}}(J_g, J_t) \tag{8}$$

The key similarity is defined as:

$$\sigma_{\text{key}}(J_g, J_t) = \frac{|\mathbb{K}_t \cap \mathbb{K}_g| + \sum_{(k_g, k_t) \in \mathbb{M}_f} \sigma_{\text{lev}}(k_g, k_t)}{|\mathbb{K}_t \cup \mathbb{K}_g|} \tag{9}$$

where $\sigma_{\text{lev}}(\cdot, \cdot)$ is the Levenshtein similarity function.

The value similarity is designed with 2 special rules to reflect the medical significance of different fields. (1) For keys containing `"id"`, any mismatch in the corresponding values is not penalized, as these identifiers do not contribute to the medical validity. (2) For keys named `"text"`, the weight of the field is increased to emphasize its importance in the overall similarity assessment.

$$\sigma_{\text{value}}(J_g, J_t) = \begin{cases} \lambda \cdot \sigma_{\text{str}}(J_g[\texttt{"text"}], J_t[\texttt{"text"}]) + (1 - \lambda) \cdot \sigma_{\text{value}}(J_g', J_t') & \text{if } \texttt{"text"} \in \mathbb{M} \\[4mm] \dfrac{\sum_{(k_g, k_t) \in \mathbb{M}} \left( \mathbb{1}_{\texttt{"id"} \in k_g} + \sigma(J_g[k_g], J_t[k_t]) \cdot \mathbb{1}_{\texttt{"id"} \notin k_g} \right) \cdot \omega(J_g[k_g])}{\sum_{k_g \in \mathbb{K}_g} \omega(J_g[k_g])} & \text{if } \texttt{"text"} \notin \mathbb{M} \end{cases} \tag{10}$$

where $J_g'$ and $J_t'$ are the dictionaries $J_g$ and $J_t$ with the `"text"` key removed. In practice, we set $\lambda = 0.5$, which is observed to ensure that the `"text"` field has a significant impact on the final score while preventing it from overshadowing the correctness judgments of other fields.

**List Similarity** is based on the best-match principle. Each element in $J_g$ is paired with its most similar element in $J_t$ using $\sigma(\cdot, \cdot)$ as the similarity function. The final score is a complexity-weighted average of these pairs, similar to the dictionary scoring logic.

**Numerical Similarity** is based on the normalized absolute difference.

$$\sigma_{\text{num}}(v_g, v_t) = \begin{cases} 1 & \text{if } v_g = 0 \text{ and } v_t = 0 \\ \max\left(1 - \dfrac{|v_g - v_t|}{\max(|v_g|, |v_t|)}, 0\right) & \text{otherwise} \end{cases} \tag{11}$$

**String Similarity** is computed using a composite score that averages Levenshtein similarity and ROUGE scores to balance different aspects of text similarity.

$$\sigma_{\text{str}}(v_g, v_t) = (\sigma_{\text{lev}}(v_g, v_t) + R_1(v_g, v_t) + R_2(v_g, v_t) + R_L(v_g, v_t)) \cdot \frac{1}{4} \tag{12}$$

A detailed example of the SimScore calculation is provided in Appendix E.

## 3.3 Training Procedure

Our training process utilizes 3 types of data: (1) the original QA pairs from the trainset, (2) the QA pairs with refined CoT, and (3) the filtered subset of CoT-augmented QA pairs.

The original QA pairs are directly constructed from the trainset, where each sample consists of a prompt and the corresponding ground-truth event. The refinement is performed with two steps: first, a strong baseline model generates an initial answer (with CoT) based on the original question from the QA pair; second, the same model is prompted to refine its reasoning steps with the ground-truth answer and SimScore. The detailed prompt for refinement can be found in Appendix C. Although the model is requested to not include any information from the ground-truth answer in its reasoning steps, there are still samples that exhibit hindsight bias, where the reasoning improperly leverages information from the ground-truth answer. These samples are filtered out to create the filtered QA pairs with refined CoT, by checking if "ground truth" or "refine" appears in the reasoning steps.

We explore and compare 2 training strategies: supervised fine-tuning (SFT) and reinforcement learning with Group Relative Policy Optimization (GRPO) (Shao et al., 2024). GRPO first prompts the model to generate multiple samples for each question, with each answer receiving its SimScore. Answers with relatively higher scores are rewarded, while those with lower scores are penalized.

## 4 Experiments

### 4.1 Experimental Settings

ClinSim is built upon Qwen3 (Yang et al., 2025). Specifically, we use Qwen3-4B for training and Qwen3-30B for parsing notes. Qwen3-30B is the fastest model with a parsing success rate over 95%, while other models are either too slow or have unsatisfactory performance. We use Qwen3-235B/8B/1.7B for comparison. Qwen3-235B is also used for CoT refinement in both generation step and refinement step.

Other models are also employed for comparison. GPT-OSS-120B (Agarwal et al., 2025), Deepseek-R1-0528-Qwen3-8B distilled from Deepseek-R1-0528 (DeepSeek-AI, 2025) are included as strong general-purpose baselines. HuatuoGPT-o1 series (Chen et al., 2024) and Baichuan-M2-32B (Dou et al., 2025) are included as state-of-the-art medical QA models. General-purpose models and medical QA models are evaluated in a few-shot manner with 1 output example for each category in the prompt.

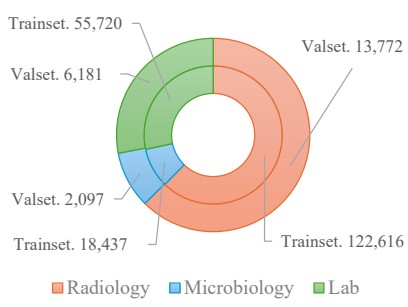

Figure 3: Proportion of Event Types.

To comply with the restrictions of MIMIC-IV and ensure patient privacy, all processes of curating, training, and evaluating are conducted locally with open-source models, and proprietary models are not utilized. For History-ExamEvent pairs generated from trajectories in ClinTrack, we truncate the input to 4,096 tokens for both training and evaluation. We split the dataset into training and evaluation sets with a ratio of 9:1, resulting in 196,773 training samples and 22,050 evaluation samples. To avoid data leakage, splits are made at the patient level, ensuring that samples from the same patient are contained within the same split. The proportion of the event types is shown in Figure 3. More implementation details can be found in Appendix A.

### 4.2 Results and Analysis

The experimental results are summarized in Table 1, with performance evaluated by SimScore. ClinSim surpasses all baseline models by at least 11.86 percentage points in overall SimScore and 11.93 percentage points in macro SimScore.

The results reveal varying difficulty across event types. Radiology and lab events are more challenging to simulate than microbiology events. This is likely because microbiology events are relatively more standardized.

| Model | Radiology | Microbiology | Lab | Overall | Macro |
|---|---|---|---|---|---|
| Qwen3-1.7B | 51.92% | 78.94% | 31.09% | 48.65% | 53.98% |
| Qwen3-1.7B$^*$ | 54.63% | 88.32% | 35.36% | 52.43% | 59.44% |
| Qwen3-4B | 54.69% | 77.93% | 37.56% | 52.10% | 56.73% |
| Qwen3-4B$^*$ | 54.17% | 67.52% | 41.69% | 51.94% | 54.46% |
| Qwen3-8B | 55.07% | 71.26% | 46.33% | 54.16% | 57.55% |
| Qwen3-8B$^*$ | 55.60% | 77.74% | 46.73% | 55.22% | 60.02% |
| DeepSeek-R1-0528-Qwen3-8B | 52.55% | 85.15% | 42.12% | 52.73% | 59.94% |
| Qwen3-32B | 56.65% | 54.43% | 49.79% | 54.52% | 53.62% |
| Qwen3-32B$^*$ | 57.00% | 76.22% | 48.69% | 56.50% | 60.64% |
| Baichuan-M2-32B$^+$ | 55.61% | 88.16% | 48.93% | 56.83% | 64.23% |
| Baichuan-M2-32B$^{+*}$ | 53.60% | 87.94% | 47.87% | 55.26% | 63.13% |
| Qwen3-235B | 55.32% | 87.05% | 47.77% | 56.22% | 63.38% |
| Qwen3-235B$^*$ | 57.01% | 72.22% | 48.90% | 56.18% | 59.38% |
| HuatuoGPT-o1-7B$^+$ | 27.86% | 56.91% | 38.26% | 33.54% | 41.01% |
| HuatuoGPT-o1-8B$^+$ | 16.83% | 75.23% | 20.66% | 23.46% | 37.57% |
| HuatuoGPT-o1-70B$^+$ | 53.23% | 72.37% | 40.68% | 51.53% | 55.43% |
| HuatuoGPT-o1-72B$^+$ | 54.96% | 81.44% | 48.13% | 55.56% | 61.51% |
| GPT-OSS-120B | 54.71% | 80.64% | 46.93% | 54.99% | 60.76% |
| ClinSim | **64.77%** | **95.29%** | **68.41%** | **68.69%** | **76.16%** |

Table 1: The performance is evaluated using SimScore. The Overall and Macro scores represent the mean SimScore across all samples and across the three categories, respectively. The best scores are highlighted in **bold**, and the second best scores are underlined. Medical QA models are denoted with **+**. Results generated by hybrid-thinking models with CoT mode disabled are denoted with $*$.

The performance of baseline models primarily depends on three factors: model size, foundation model quality, and medical specialization. The Spearman correlation coefficient between model size and overall score is 0.6591 ($p$-value: 0.0029), and between model size and macro score is 0.5254 ($p$-value: 0.0252), indicating a statistically significant positive correlation. HuatuoGPT-o1-7B and HuatuoGPT-o1-8B are based on Qwen2.5 (Yang et al., 2024) and Llama3.1 (Dubey et al., 2024), and their relatively poor performance may be attributed to the smaller model sizes and less advanced foundation models. Flagship medical QA models display competitive performance, suggesting the benefits of medical specialization. Baichuan-M2-32B achieves the best overall and macro SimScore among baselines, and HuatuoGPT-o1-72B also shows strong results.

For models with hybrid-thinking mode, activating chain-of-thought (CoT) reasoning can occasionally degrade performance (Qwen3-1.7B/8B/32B). We design an ablation study using models trained exclusively for CoT and their hybrid-thinking counterparts with thinking mode enabled or disabled, as shown in Table 2. We hypothesize that due to the novelty of the examination feedback simulation task, the ordinary CoT reasoning, heavily trained on domains like mathematics, logic, programming and conventional medical QA, may not align well with the nuanced patterns required for this specific task. This misalignment can lead to a suboptimal reasoning trajectory, resulting in performance even worse than direct generation without CoT.

| Model | Overall | Macro |
|---|---|---|
| Qwen3-4B | 52.10% | 56.73% |
| Qwen3-4B$^*$ | 51.94% | 54.46% |
| Qwen3-4B-Thinking | 47.60% | 54.81% |
| Qwen3-235B | 56.22% | 63.38% |
| Qwen3-235B$^*$ | 56.18% | 59.38% |
| Qwen3-235B-Thinking | 48.04% | 55.82% |

Table 2: Ablation study on CoT. Models trained exclusively for CoT thinking are denoted with -Thinking. Non-thinking mode results of hybrid-thinking models are denoted with $*$.

The results in Table 3 offer a comprehensive view of the interaction between data quality, quantity, and training strategies: (1) Any SFT-based method significantly outperforms the few-shot baseline (52.10%), demonstrating that SFT is an effective approach to specialize the model for this task. (2) SFT on QA pairs (66.34%) outperforms SFT on CoT data (64.73%), likely because CoT data generated by existing models suffers from the

| QA | CoT | FCoT | GRPO | Radiology | Microbiology | Lab | Overall | Macro |
|---|---|---|---|---|---|---|---|---|
| | | | | 54.69% | 77.93% | 37.56% | 52.10% | 56.73% |
| ✓ | | | | 61.00% | 94.85% | **68.55%** | 66.34% | 74.80% |
| | ✓ | | | 61.86% | 94.10% | 61.15% | 64.73% | 72.37% |
| | | ✓ | | 59.98% | 93.74% | 53.18% | 61.28% | 68.97% |
| | | | ✓ | 60.10% | 92.05% | 51.38% | 60.69% | 67.84% |
| ✓ | | ✓ | | 62.10% | 94.49% | 66.62% | 66.45% | 74.40% |
| | ✓ | ✓ | | 62.51% | 94.83% | 66.67% | 66.75% | 74.67% |
| | ✓ | ✓ | ✓ | **64.77%** | **95.29%** | 68.41% | **68.69%** | **76.16%** |

Table 3: Experimental results of different training methods on Qwen3-4B. The best scores are highlighted in **bold**. QA denotes SFT with QA pairs, CoT denotes SFT with CoT-augmented samples, FCoT denotes SFT with filtered CoT-augmented samples, and GRPO denotes GRPO fine-tuning. The order of applying different training methods is from left to right.

| Model | Levenshtein | ROUGE-1 | ROUGE-2 | ROUGE-L | SimScore |
|---|---|---|---|---|---|
| Qwen3-1.7B | 0.5152 | 0.4229 | 0.2769 | 0.3733 | 0.4865 |
| Qwen3-235B | 0.5469 | 0.4547 | 0.3040 | 0.3983 | 0.5622 |
| $\Delta$ | +0.0317 | +0.0318 | +0.0271 | +0.0250 | **+0.0757** |

Table 4: Sensitivity analysis comparing SimScore with other widely used metrics. The improvement from Qwen3-1.7B to Qwen3-235B is more pronounced in SimScore. For alignment with other metrics, decimals instead of percentages are used. The largest improvement is highlighted in **bold**. The results are the average scores of all samples (Overall).

aforementioned hindsight bias and misalignment. (3) The high-quality but low-volume FCoT data (61.28%) performs worse than the noisy but high-volume CoT data. However, FCoT serves as a crucial supplement to raise the performance ceiling when combined, as evidenced by the improvement of CoT+FCoT (66.75%) over CoT alone. (4) The combination of CoT and FCoT followed by GRPO achieves the best performance (68.69%), where GRPO acts as a final stage to further polish the model's reasoning capabilities.

## 4.3 STUDIES ON SIMSCORE DESIGN

**Component Ablation Studies:** Results generated by Qwen3-4B are employed for the ablation study on different components of SimScore, as shown in Figure 4. The id skipping mechanism slightly increases the score as it prevents the model from being unfairly penalized for mismatches in non-clinical fields. The text weight adjustment substantially lowers the score to reflect the model's performance on report text simulation more fairly and acutely. The complexity weight adjustment leads to a decrease in score, which is attributable to the greater penalty applied to complex fields that are generally harder to simulate accurately.

**Overall Sensitivity Studies:** Table 4 presents a sensitivity analysis comparing SimScore with other metrics. From Qwen3-1.7B to significantly better-performing Qwen3-235B, SimScore shows a more pronounced improvement compared to other metrics, suggesting its superior effectiveness in capturing quality differences.

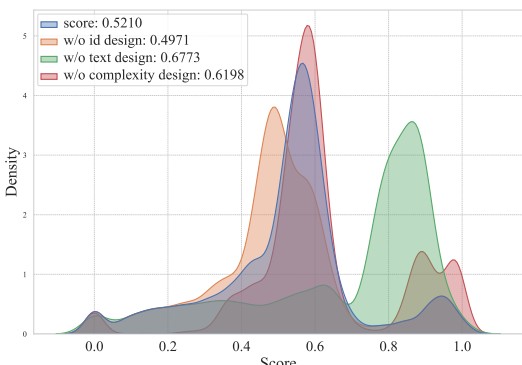

Figure 4: SimScore distributions for Qwen3-4B under different component settings. The id skipping mechanism increases the score, while text weight and complexity weight adjustments decrease the score.

**Focused Human Review on SimScore:** To assess the alignment of the proposed SimScore with expert judgment, a focused human review is conducted with two independent medical experts on 36 randomly selected samples. Each sample includes a clinical context, the examination order, and two simulated results generated by ClinSim and Qwen3-4B, respectively. To ensure an unbiased evaluation, the results are anonymized and presented in a random order, with chain-of-thought reasoning masked to prevent influence on the experts' assessments. The results are illustrated in Figures 6 and 5. The two experts exhibit high agreement on the relative quality of the models. Conflicting judgments, where one expert prefers ClinSim while the other prefers Qwen3-4B, occur in only 3 cases. SimScore shows exceptional alignment with this human consensus: it correctly identifies the winner with 90.48% accuracy on samples where experts reach a strong consensus. Furthermore, SimScore effectively mirrors human confidence: the average score margin for samples with clear human preference is 3.99 times larger (0.3112 vs. 0.0780) than for ambiguous samples. Finally, a strong linear correlation exists between the aggregated Human Consensus Score and the SimScore difference (Pearson $r = 0.61$, $p < 0.001$), confirming that SimScore scales proportionally to the degree of human certainty.

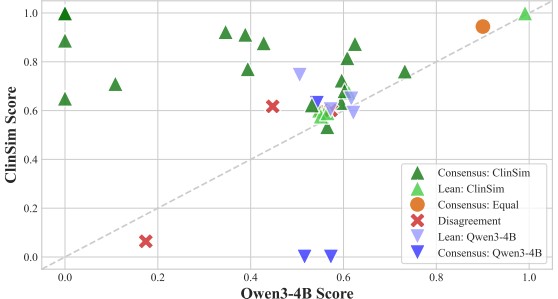 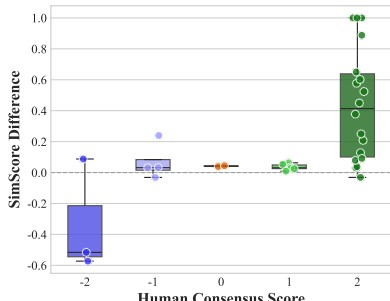

Figure 5: Alignment between SimScore and human consensus. Triangles (▲, ▼) represent samples where experts reached a consensus on the winner, while circles (●) represent consensus on a tie. Darker colors indicate stronger human consensus. Samples with human disagreement are marked with crosses (✖). SimScore aligns well with expert judgments, with disagreements clustering near the diagonal.

Figure 6: Correlation between Human Consensus Score and SimScore difference. The X-axis represents aggregated expert votes (from −2: unanimous preference for Baseline, to +2: unanimous preference for ClinSim). SimScore is calibrated to human certainty, yielding larger score margins when experts strongly agree.

# 5 DISCUSSION

## 5.1 CONCLUSION

In this work, we address a fundamental challenge in developing realistic clinical simulation environments: the limitation of clinical trajectory uniqueness, which arises from the static nature of existing datasets. We introduce a comprehensive framework for simulating patient examination feedback, encompassing the task formulation, the large-scale ClinTrack-MIMIC dataset, and the tailored structure-aware evaluation metric SimScore.

Our empirical evaluation reveals that both general-purpose and existing specialized models fail to perform satisfactorily on this task, while the proposed ClinSim model (4B parameters) shows promising results, significantly outperforming larger models (up to 235B parameters) by over 10 percentage points on SimScore, thus highlighting the potential of a specialized LLM in this task.

This work represents the first attempt to systematically investigate the under-explored problem of patient examination feedback simulation, laying the foundation for creating dynamic longitudinal diagnosis-and-treatment simulation frameworks for training and evaluating advanced clinical decision-making LLMs.

## 5.2 FUTURE WORK

While this study utilizes the MIMIC-IV dataset to ensure reproducibility and focuses on structured inpatient data, our framework is intentionally designed for extensibility across diverse clinical scenarios. The proposed JSON-based format and the structure-aware SimScore are modality-agnostic. Specifically, new modalities (e.g., medical images) or datasets from different clinical centers can be integrated into ClinTrack, and SimScore supports extension by appending task-specific metrics (e.g., image similarity) as additional leaf nodes in the evaluation tree. It is also promising to leverage information extraction techniques to parse unstructured data (e.g., ECG records) from discharge notes and extend the simulation scope to outpatient settings. Furthermore, the ClinTrack dataset is expected to serve as a valuable resource for various sequential clinical tasks, including pretraining medical LLMs through next-event prediction.

Beyond expanding the scope, exploring the utility of the simulator in interactive environments is a compelling direction. As simulation frameworks have been widely adopted in other domains not only for evaluation but also for training, such as simulated spatial interaction for autonomous driving (Wang et al., 2024a; Zhao et al., 2025) and simulated search feedback for LLM tool use (Sun et al., 2025a), we argue that the potential of examination feedback simulation for medical agent training is underestimated and deserves dedicated research.

Finally, as the fidelity of simulation improves, we envision adopting a Turing test-style evaluation as the ultimate metric to assess the indistinguishability of simulated clinical feedback from real-world medical records.

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

# A  IMPLEMENTATION DETAILS

During training, our models are all initialized from Qwen3-4B. The training process is conducted with TRL (von Werra et al., 2020) on 8 NVIDIA L20 GPUs, and 2 settings are explored:

**For SFT**, we train the model with a learning rate of 1e-5 and a weight decay of 1e-4 using the AdamW (Loshchilov & Hutter, 2017) optimizer. For the original QA pairs from the trainset and the QA pairs with refined CoT, the model is trained for 1 epoch, while for the filtered CoT-augmented QA pairs, it is trained for 3 epochs. A cosine learning rate scheduler is employed with a warmup ratio of 0.05. We set the per-device batch size to 1 and utilize 64 gradient accumulation steps. To manage memory constraints, we use bfloat16 precision and DeepSpeed (Rasley et al., 2020) with ZeRO-3 optimization.

**For GRPO**, we train the model using the AdamW optimizer with a learning rate of 1e-6. For regularization, $\beta$ is set to 1e-3 and $\epsilon$ is set to 0.2. The scheduler, batch size, gradient accumulation steps, precision and DeepSpeed share the same configuration as in SFT. Rollout is set to 4 per sample. 2 GPUs are used for rollout with vLLM (Kwon et al., 2023) and 6 GPUs are used for training. During GRPO-only training, the model is trained for 500 steps, leading to a total of 192,000 rollout samples being processed. When combined with SFT, GRPO training is performed for 200 steps as it saturates more quickly compared to GRPO-only training, resulting in 76,800 rollout samples.

# B  DISEASE CATEGORY DISTRIBUTION

The disease category distribution in ClinTrack is illustrated in Figure 7.

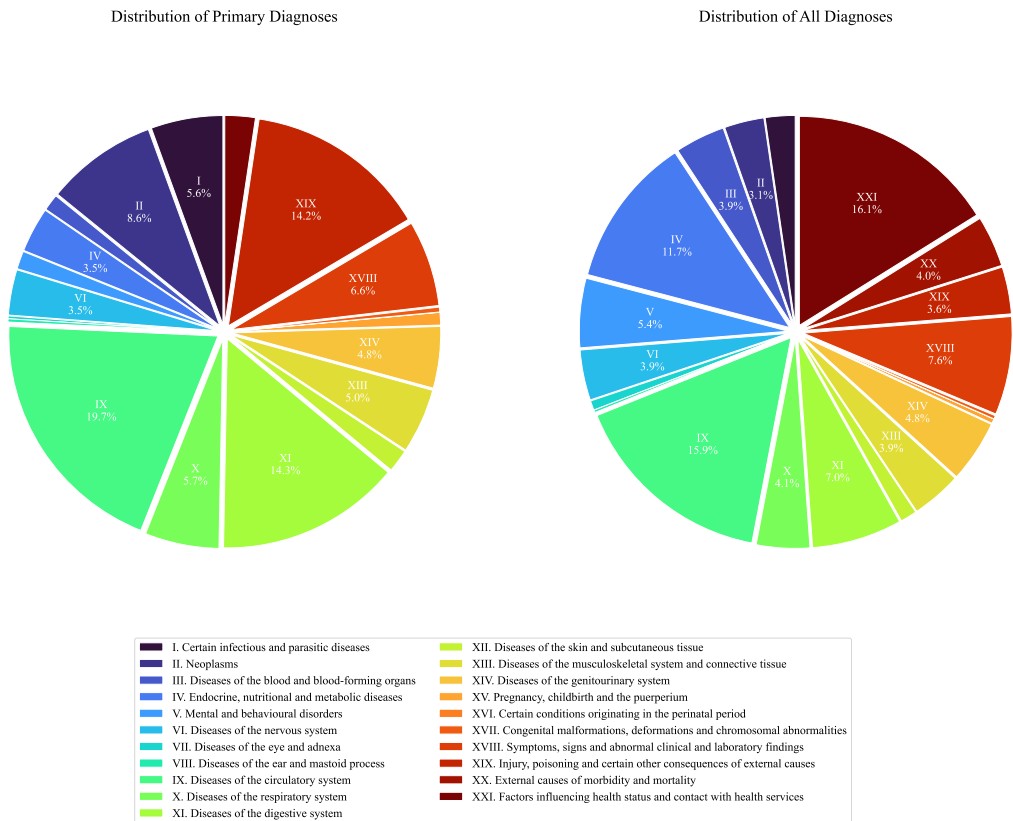

Figure 7: Primary Disease (the first diagnosis of each admission) Category Distribution (left) and Overall Disease Category Distribution (right) in ClinTrack

## C  PROMPT DESIGN

The detailed prompts used in our experiments with ClinSim are provided in this section. The prompt for note extraction is shown in Table 5. The prompt for report extraction is shown in Table 6. The prompt for examination feedback simulation is shown in Table 7. The prompt for CoT refinement is shown in Table 8.

| Role | Content |
|------|---------|
| System | You are a medical assistant.  You are responsible for parsing the following discharge note and extracting the relevant information in JSON format.  You should strictly copy the information from the discharge note and paste it into the corresponding fields.  If any information is missing, please leave the field empty by filling it with null.  The JSON should be as follows:  { "Admission Info":  { "Allergies":  "text", "Chief Complaint":  "text", "History of Present Illness":  "text", "Past Medical History":  "text", "Social History":  "text", "Family History":  "text", "Admission Physical Exam":  "text", "Medications on Admission":  "text", }, "Progress Info": { "Major Surgical or Invasive Procedure":  "text", "Pertinent Results":  "text", "Brief Hospital Course": "text", }, "Discharge Info":  { "Discharge Condition": "text", "Discharge Physical Exam":  "text", "Discharge Instructions":  "text", "Discharge Diagnosis":  "text", "Discharge Medications":  "text", "Followup Instructions": "text" } } Note:  all "text" should be replaced with the actual text from the discharge note in plain text format, the answer should not include any type of other elements like "```json". |
| User | \<few-shot discharge note example\> |
| Assistant | \<few-shot parsed discharge note example\> |
| User | \<discharge note to be parsed\> |

Table 5: Prompt for Note Extraction

| Role | Content |
|------|---------|
| System | You need to determine what type of imaging modality the radiology note below is from:  ["X-ray", "CT", "MRI", "Ultrasound", "Fluoroscopy", "Others"].  You should only provide one of the above options as your answer.  Note that Computed Tomography is also known as CT, and Magnetic Resonance Imaging is also known as MRI. If you are not sure or the modality is not listed, please choose Others. Your answer should be one of the following:  ["X-ray", "CT", "MRI", "Ultrasound", "Fluoroscopy", "Others"]. Please do not include any other content in your answer. |
| User | \<few-shot radiology report example\> |
| Assistant | \<few-shot parsed radiology report example\> |
| User | \<radiology report to be parsed\> |

Table 6: Prompt for Report Extraction

| Role | Content |
|---|---|
| System | You are a medical prediction assistant tasked with forecasting the potential results of a patient's subsequent medical examination based on their existing data. Representative examples are provided below. Information to Predict: <radiology event few-shot query example> Your response should be: <radiology event few-shot response example> Information to Predict: <microbiology event few-shot query example> Your response should be: <microbiology event few-shot response example> Information to Predict: <lab event few-shot query example> Your response should be: <lab event few-shot response example> |
| User | Preliminary Information: <preliminary information> Information to Predict: <event query with time and type> |

Table 7: Prompt for Examination Feedback Simulation

| Role | Content |
|---|---|
| System | You are to act as a Chain-of-Thought (CoT) rewriting expert. You will be provided with an existing conversation record, which includes the following components: A System Prompt that defines the task. A user's Question. A Ground Truth Answer, which is the correct answer to the question, but without a chain of thought. An Existing Answer, which consists of a chain of thought and a final answer. A Score for the existing answer. This score serves as a reference to help you determine the extent to which the original chain of thought should be retained or revised. Before generating the final, refined chain of thought, you may engage in your own preliminary reasoning or analysis. After completing this process, you must present the refined chain of thought in your final output, enclosed in the following format:<cot> [Your refined chain of thought here] </cot> Important Note: The chain of thought must present a meticulous, step-by-step reasoning process. The analysis must be sufficiently rigorous to ensure its conclusion closely aligns with (without necessarily being identical to) the provided ground truth. It is crucial to avoid revealing the final answer prematurely. Furthermore, any form of reverse causality (e.g., stating, "Because the ground truth answer is ..., therefore...") is strictly prohibited. |
| User | System Prompt of the Task: <system prompt>; User's Question: <user question>; Ground Truth Answer: <ground truth answer>; Existing Answer: <existing answer>; Score: <accuracy score>. |

Table 8: Prompt for CoT Refinement

# D    DETAILED DATA STRUCTURE DESCRIPTION

MIMIC-IV and its derivatives have strict data usage policies that prevent us from sharing actual patient data. Figure 8 illustrates an example data structure from the ClinTrack dataset.

```
{
    "subject_id": ...,
    "gender": ...,
    "birth": ...,
    "dod": ...,
    "admission": [
        {
            "hadm_id": ...,
            "language": ...,
            "marital_status": ...,
            "race": ...,
            "diagnosis_list": [
                {
                    "seq_num": ...,
                    "icd_code": ...,
                    "icd_version": ...,
                    "long_title": ...
                },
                ...
            ],
            "discharge_note": ...,
            "admission_info": {
                "allergies": ...,
                "chief_complaint": ...,
                "history_of_present_illness": ...,
                "past_medical_history": ...,
                "social_history": ...,
                "family_history": ...,
                "admission_physical_exam": ...,
                "medications_on_admission": ...
            },
            "progress_info": {
                "major_surgical_or_invasive_procedure": ...,
                "pertinent_results": ...,
                "brief_hospital_course": ...
            },
            "discharge_info": {
                "discharge_condition": ...,
                "discharge_physical_exam": ...,
                "discharge_instructions": ...,
                "discharge_diagnosis": ...,
                "discharge_medications": ...,
                "followup_instructions": ...
            },
            "event_list": [
                {
                    "event_time": ...,
                    "event_type": "TransferEvent",
                    "transfer_location": ...
                },
                {
                    "event_time": ...,
                    "event_type": "RadiologyEvent",
                    "note_id": ...,
                    "note_type": ...,
                    "note_seq": ...,
                    "text": ...,
```

```
                        "modality": "X-Ray"
                    },
                    {
                        "event_time": ...,
                        "event_type": "AdmissionEvent",
                        "admission_type": ...,
                        "admission_location": ...
                    },
                    {
                        "event_time": ...,
                        "event_type": "ServiceEvent",
                        "prev_service": ...,
                        "curr_service": ...
                    },
                    {
                        "event_time": ...,
                        "event_type": "TransferEvent",
                        "transfer_location": ...
                    },
                    {
                        "event_time": ...,
                        "event_type": "MicrobiologyEvent",
                        "spec_type_desc": ...,
                        "test_name": ...,
                        "microbiology_list": [
                            {
                                "micro_event_id": ...,
                                "test_seq": ...,
                                "org_itemid": ...,
                                "org_name": ...,
                                "isolate_num": ...,
                                "quantity": ...,
                                "ab_itemid": ...,
                                "ab_name": ...,
                                "dilution_text": ...,
                                "dilution_comparison": ...,
                                "dilution_value": ...,
                                "interpretation": ...,
                                "comments": ...
                            },
                            ...
                        ]
                    },
                    {
                        "event_time": ...,
                        "event_type": "LabEvent",
                        "specimen_id": ...,
                        "fluid": ...,
                        "category": ...,
                        "lab_list": [
                            {
                                "lab_event_id": ...,
                                "itemid": ...,
                                "ref_range_lower": ...,
                                "ref_range_upper": ...,
                                "flag": ...,
                                "value": ...,
                                "valuenum": ...,
                                "valueuom": ...,
                                "priority": ...,
                                "comments": ...,
                                "label": ...
                            },
```

```
                         ...
                    ]
               },
               {
                    "event_time": ...,
                    "event_type": "EmarEvent",
                    "emar_list": [
                         {
                              "emar_id": ...,
                              "emar_seq": ...,
                              "medication": ...,
                              "event_txt": ...
                         },
                         ...
                    ]
               },
               {
                    "event_time": ...,
                    "event_type": "ProcedureEvent",
                    "seq_num": ...,
                    "icd_code": ...,
                    "icd_version": ...,
                    "long_title": ...
               },
               ...
          ]
     }
    ]
}
```

Figure 8: Example Data Entry in ClinTrack

## E    EXAMPLE SIMSCORE CALCULATION

In this section, we provide a detailed example of calculating the SimScore between a ground truth JSON structure and a predicted JSON structure. Let the ground truth structure be as shown in Figure 9.

```
{
    "key_id": 12345678,
    "key_1": "str value 1",
    "key_2": "str value 2",
    "key_3": "str value 3",
    "key_4": "str value 4",
    "key_5": 5.0,
    "list_field": [
        {
            "sub_1_key_id": 12345678,
            "sub_1_key_1": "str value 2",
            "sub_1_key_2": 2.0,
            "sub_1_key_3": 3.0,
            "sub_1_key_4": 4.0
        },
        {
            "sub_2_key_id": 12345678,
            "sub_2_key_1": "str value 3",
            "sub_2_key_2": 5.0,
            "sub_2_key_3": 6.0,
            "sub_2_key_4": 7.0
```

```
        },
        {
            "sub_3_key_id": 12345678,
            "sub_3_key_1": "str value 4",
            "sub_3_key_2": 8.0,
            "sub_3_key_3": 9.0,
            "sub_3_key_4": 10.0
        }
    ]
}
```

Figure 9: Example Ground Truth

Let a predicted result be as shown in Figure 10.

```
{
    "key_id": 87654321,
    "key_1": "str value 1",
    "key_2": "str value 2",
    "key_3": "str value 3",
    "key_4": "str value 4.1",
    "key_5": 5.0,
    "list_field": [
        {
            "sub_1_key_id": 87654321,
            "sub_1_key_1": "str value 2",
            "sub_1_key_2": 2.1,
            "sub_1_key_3": 3.1
        },
        {
            "sub_3_key_id": 87654321,
            "sub_3_key_1": "str value 4",
            "sub_3_key_2": 8.1,
            "sub_3_key_3": 9.1
        }
    ]
}
```

Figure 10: Predicted Example

### E.1 STEP 1: COMPLEXITY WEIGHTING ANALYSIS

First, we calculate the complexity weight $\omega(v)$ for the ground truth structure $J_g$ according to Eq. 7.

**Weights of Individual List Elements**   The "list_field" in $J_g$ contains three dictionaries: $G_1$, $G_2$, and $G_3$. Each dictionary contains 5 fields (keys). Since all values within these dictionaries are primitive types, their individual weights are 1. Thus, the weight for each element is:

$$\omega(G_1) = \omega(G_2) = \omega(G_3) = \sum_{k \in \mathcal{K}_{G_i}} 1 = 5 \tag{13}$$

**Weight of the List**   The weight of the list $J_g["list\_field"]$ is the sum of the weights of its elements:

$$\omega(J_g["list\_field"]) = 5 + 5 + 5 = 15 \tag{14}$$

**Total Weight of $J_g$**   The top-level dictionary contains 6 primitive fields (weight = 1 each) and the "list_field" (weight = 15). The total complexity weight is:

$$\omega(J_g) = 6 \times 1 + 15 = 21 \tag{15}$$

### E.2 STEP 2: SIMILARITY OF THE LIST FIELD

We compute $\sigma_{\text{list}}(J_g[\texttt{"list\_field"}], J_t[\texttt{"list\_field"}])$. Based on the best-match principle, we align the elements:

- Pair 1: $G_1$ matches $P_1$.
- Pair 2: $G_2$ has no match in $J_t$ (Missing).
- Pair 3: $G_3$ matches $P_2$.

#### E.2.1 CALCULATION FOR PAIR 1: $G_1, P_1$

**Key Similarity:** The key sets are $\mathcal{K}_{G_1}$ (5 keys) and $\mathcal{K}_{P_1}$ (4 keys). The prediction is missing the key $\texttt{"sub\_1\_key\_4"}$.

$$\sigma_{\text{key}}(G_1, P_1) = \frac{|\mathcal{K}_{G_1} \cap \mathcal{K}_{P_1}|}{|\mathcal{K}_{G_1} \cup \mathcal{K}_{P_1}|} = \frac{4}{5} = 0.8 \tag{16}$$

**Value Similarity:** We sum the similarity of values for the intersection of keys. Notable calculations include:

1. $\texttt{"sub\_1\_key\_id"}$: Contains substring "id". Score = 1.0.
2. $\texttt{"sub\_1\_key\_1"}$: Exact match. Score = 1.0.
3. $\texttt{"sub\_1\_key\_2"}$ ($v_g = 2, v_t = 2.1$):

$$\sigma_{\text{num}} = 1 - \frac{|2 - 2.1|}{2.1} \approx 0.9524$$

4. $\texttt{"sub\_1\_key\_3"}$ ($v_g = 3, v_t = 3.1$):

$$\sigma_{\text{num}} = 1 - \frac{|3 - 3.1|}{3.1} \approx 0.9677$$

5. $\texttt{"sub\_1\_key\_4"}$: Missing in prediction.

The $\sigma_{\text{value}}$ is thus:

$$\sigma_{\text{value}}(G_1, P_1) = \frac{1.0 + 1.0 + 0.9524 + 0.9677}{5} = \frac{3.9201}{5} \approx 0.7840 \tag{17}$$

The $\sigma_{\text{dict}}$ for this pair is:

$$\sigma_{\text{dict}}(G_1, P_1) = \sigma_{\text{key}}(G_1, P_1) \times \sigma_{\text{value}}(G_1, P_1) = 0.8 \times 0.7840 = 0.6272 \tag{18}$$

#### E.2.2 CALCULATION FOR PAIR 3: $G_3, P_2$

**Key Similarity:** Same as Pair 1, $\sigma_{\text{key}} = 0.8$ (missing $\texttt{"sub\_3\_key\_4"}$).

**Value Similarity:**

1. $\texttt{"sub\_3\_key\_id"}$: Contains substring "id". Score = 1.0.
2. $\texttt{"sub\_3\_key\_1"}$: Exact string match. Score = 1.0.
3. $\texttt{"sub\_3\_key\_2"}$ ($v_g = 8, v_t = 8.1$):

$$\sigma_{\text{num}} = 1 - \frac{|8 - 8.1|}{8.1} \approx 0.9877$$

4. $\texttt{"sub\_3\_key\_3"}$ ($v_g = 9, v_t = 9.1$):

$$\sigma_{\text{num}} = 1 - \frac{|9 - 9.1|}{9.1} \approx 0.9890$$

5. $\texttt{"sub\_3\_key\_4"}$: Missing in prediction.

Sum of value scores: $1.0 + 1.0 + 0.9877 + 0.9890 = 3.9767$.

$$\sigma_{\text{dict}}(G_3, P_2) = 0.8 \times \frac{3.9767}{5} \approx 0.6363 \tag{19}$$

### E.2.3 Aggregated List Score

The list similarity is the weighted average of the pair scores against the ground truth weights:

$$
\begin{aligned}
\sigma_{\text{list}} &= \frac{\sigma(G_1, P_1) \cdot \omega(G_1) + \sigma(G_2, \text{None}) \cdot \omega(G_2) + \sigma(G_3, P_2) \cdot \omega(G_3)}{\omega(J_g[\texttt{"list\_field"}])} \\
&= \frac{(0.6272 \times 5) + (0 \times 5) + (0.6363 \times 5)}{15} \\
&\approx 0.4212
\end{aligned}
\tag{20}
$$

### E.3 Step 3: Top-Level Similarity and Final Score

Finally, we compute the similarity for the root object.

**Key Similarity:** The keys of $J_g$ and $J_t$ are identical.

$$
\sigma_{\text{key}}(J_g, J_t) = 1.0
\tag{21}
$$

**Value Similarity:** We sum the similarity contributions of the top-level fields, weighted by their complexity:

1. `"key_id"`: Contains substring "id". Score = 1.0.

2. `"key_1"`, `"key_2"`, `"key_3"`, `"key_5"`: exact matches.

$$
\text{Score} = 4 \times 1.0 = 4.0
$$

3. `"key_4"`: `"str value 4"` vs. `"str value 4.1"`, the average of ROUGE-1, ROUGE-2, ROUGE-L and Levenshtein similarity:

$$
\text{Score} = (0.8571 + 0.8000 + 0.8571 + 0.8462) \cdot \frac{1}{4} \approx 0.8401
$$

4. `"list_field"`: Uses $\sigma_{\text{list}}$ calculated above.

$$
\text{Score} = 0.4212 \times 15 = 6.3180
$$

The aggregated value similarity is:

$$
\sigma_{\text{value}}(J_g, J_t) = \frac{1.0 + 4.0 + 0.8401 + 6.3180}{21} \approx 0.5790
\tag{22}
$$

**Final SimScore:** The final structural similarity score is the product of the top-level key and value similarities:

$$
s = \sigma(J_g, J_t) = 1.0 \times 0.5790 = 0.5790
\tag{23}
$$

Consider a better prediction shown in Figure 11.

```
{
    "key_id": 87654321,
    "key_1": "str value 1",
    "key_2": "str value 2",
    "key_3": "str value 3",
    "key_4": "str value 4.1",
    "key_5": 5.0,
    "list_field": [
        {
            "sub_1_key_id": 87654321,
            "sub_1_key_1": "str value 2",
            "sub_1_key_2": 2.1,
            "sub_1_key_3": 3.0,
            "sub_1_key_4": 4.0
```

```
        },
        {
            "sub_2_key_id": 87654321,
            "sub_2_key_1": "str value 3",
            "sub_2_key_2": 5.1,
            "sub_2_key_3": 6.5,
            "sub_2_key_4": 7.2
        },
        {
            "sub_3_key_id": 87654321,
            "sub_3_key_1": "str value 4",
            "sub_3_key_2": 8.1,
            "sub_3_key_3": 9.1,
            "sub_3_key_4": 10.0
        }
    ]
}
```

Figure 11: Another Better-Matching Predicted Example

The calculated SimScore for this better prediction is approximately 0.9831.

# F  HUMAN AUDIT OF THE PARSING PIPELINE

To quantify the potential label noise introduced by our LLM-based parsing pipeline, we conducted a comprehensive human audit covering both the discharge note extraction and radiology modality classification tasks.

## F.1  AUDIT OF DISCHARGE NOTE EXTRACTION

For the structured extraction of discharge notes, we randomly sampled 50 cases from our dataset and manually annotated the ground truth JSONs. The LLM-generated labels were then compared against these human-annotated labels. We measured the agreement using ROUGE (R-1, R-2, R-L) and Levenshtein similarity. The results, summarized in Table 9, show a very high agreement between the LLM-generated labels and the human audit. The high scores confirm that our extraction pipeline is highly accurate, ensuring that the structured data faithfully represents the original clinical notes.

## F.2  AUDIT OF RADIOLOGY MODALITY CLASSIFICATION

For the radiology modality classification task, we performed a stratified sampling of 125 reports to ensure balanced coverage across all categories. Specifically, we randomly selected 25 samples for each of the five defined modalities: X-ray, CT, Ultrasound, MRI, and Fluoroscopy.

Given that radiology reports typically contain explicit modality keywords in their descriptions, this classification task is inherently easy. The human audit confirmed this observation, yielding an accuracy of 100% across all 125 sampled cases. This perfect alignment indicates that the modality labels used in our dataset are highly reliable, with negligible risk of label noise affecting downstream analyses.

| Metric | Score |
|---|---|
| ROUGE-1 | 97.29% |
| ROUGE-2 | 90.48% |
| ROUGE-L | 97.29% |
| Levenshtein Similarity | 96.92% |

Table 9: Human Audit of LLM-Generated Label Accuracy for Discharge Note Extraction

# G  DEMOGRAPHIC SUBGROUP ANALYSIS

To ensure the robustness of ClinSim across different demographic subgroups, we conducted a comprehensive evaluation. This assessment covers major demographic dimensions, including race, age group, and gender. The results demonstrate consistent performance across all subgroups, supporting the model's suitability for deployment in heterogeneous clinical populations.

## G.1  RACE-BASED PERFORMANCE

ClinSim shows stable behavior across racial groups, with mean performance scores remaining within a narrow range. Although the White and Black cohorts constitute the majority of the dataset, smaller groups such as Asian and Hispanic populations exhibit consistent performance outcomes, suggesting no observable algorithmic bias related to racial composition.

| Race | Count | Overall | Overall STD | Macro | Macro STD |
|------|-------|---------|-------------|-------|-----------|
| Asian | 958 | 70.14% | 18.36% | 77.13% | 17.87% |
| Black | 2,314 | 68.75% | 17.31% | 76.40% | 16.78% |
| Hispanic | 1,009 | 69.20% | 18.34% | 76.06% | 16.64% |
| White | 15,663 | 68.52% | 17.20% | 76.16% | 16.56% |
| Unknown | 2,106 | 68.97% | 18.55% | 75.44% | 16.81% |

Table 10: Performance Across Race Subgroups

## G.2  AGE GROUP ANALYSIS

ClinSim maintains consistent performance across various age groups, with only a slight decrease (-1.96 percentage points compared to mean performance among all age groups) observed in geriatric demographics. Despite this, the variations remain minimal, underscoring the model's reliability across age-related clinical heterogeneity.

| Age Group | Count | Overall | Overall STD | Macro | Macro STD |
|-----------|-------|---------|-------------|-------|-----------|
| 18-39 (Young Adult) | 3,191 | 69.86% | 18.29% | 77.14% | 16.53% |
| 40-64 (Middle Age) | 8,437 | 69.60% | 18.33% | 76.41% | 16.86% |
| 65-79 (Senior) | 5,857 | 68.27% | 17.06% | 75.64% | 15.86% |
| 80+ (Geriatric) | 4,565 | 66.73% | 15.38% | 75.48% | 17.45% |

Table 11: Performance Across Age Groups

## G.3  GENDER COMPARISON

Gender-based comparisons reveal nearly identical performance distributions between male and female cohorts. The similarity in all reported statistics suggests that ClinSim's predictions are not significantly affected by gender, reinforcing the fairness and stability of the model.

| Gender | Count | Overall | Overall STD | Macro | Macro STD |
|--------|-------|---------|-------------|-------|-----------|
| M | 11,104 | 68.78% | 17.61% | 76.01% | 16.72% |
| F | 10,946 | 68.60% | 17.28% | 76.30% | 16.64% |

Table 12: Performance Across Gender

# H ABLATION STUDY ON TIMESTAMP SENSITIVITY

To investigate the role of precise timestamps in our simulation framework, we conducted an ablation study where the target timestamp was either removed or perturbed. We evaluated the model under the following settings:

- **Timestamp Removed**: The target timestamp is omitted from the input.

- **Timestamp Perturbed**: The timestamp is shifted (1) backward to the time of the last observed event, or (2) forward by 1 day, 7 days, and 30 days, respectively.

For a fair comparison focused on content generation, the `event_time` field was excluded from the SimScore calculation (i.e., value similarity for this field was set to 1.0).

The results are presented in Table 13. We observe a consistent trend: while the model maintains a high baseline performance, larger temporal perturbations lead to progressively larger decreases in performance.

| Setting | Radiology | Microbiology | Lab | Overall | Macro |
|---|---|---|---|---|---|
| With Timestamp | 64.74 | 95.19 | 68.78 | 68.77 | 76.24 |
| Timestamp Removed | 64.62 | 95.09 | 68.48 | 68.60 | 76.06 |
| Timestamp Perturbed, backward | 64.59 | 95.13 | 68.10 | 68.48 | 75.94 |
| Timestamp Perturbed, forward (1 day) | 64.39 | 94.87 | 68.18 | 68.35 | 75.82 |
| Timestamp Perturbed, forward (7 days) | 64.09 | 94.68 | 67.86 | 68.06 | 75.54 |
| Timestamp Perturbed, forward (30 days) | 64.12 | 94.59 | 67.62 | 68.00 | 75.45 |

Table 13: Ablation Study on Timestamp Sensitivity. Performance is reported in SimScore (%).

The results indicate that the clinical history is the primary driver for structural correctness and event causality, underscoring the model's robustness. Meanwhile, the downward trend with larger perturbations confirms that the timestamp serves as an essential modulator for fine-grained physiological values, suggesting that the model effectively leverages temporal information to refine its predictions.

To further validate that the model's sensitivity aligns with clinical reality, we quantified the intrinsic temporal dependency within the dataset. We performed an analysis on a cohort of 3,000 randomly sampled patients by computing the Pearson correlation between the time gap (in hours) and the content similarity (SimScore) of the first and last events of the same type.

The results are summarized in Table 14. We observe a statistically significant negative correlation across all event types ($p < 0.001$). The extremely low p-values confirm the certainty of patient state drift over time, while the magnitude of the correlation coefficients reflects the gradual nature of this physiological evolution. This intrinsic data characteristic implies that patient states evolve continuously rather than abruptly, which explains the progressive performance decay observed in our ablation study.

| Event Type | Count | Pearson R | P-Value | Slope |
|---|---|---|---|---|
| LabEvent | 4,094 | -0.1016 | $7.24 \times 10^{-11}$ | $-4.86 \times 10^{-5}$ |
| MicrobiologyEvent | 1,129 | -0.1150 | $1.08 \times 10^{-4}$ | $-2.48 \times 10^{-5}$ |
| RadiologyEvent | 2,305 | -0.1783 | $6.42 \times 10^{-18}$ | $-5.05 \times 10^{-5}$ |

Table 14: Analysis of temporal dependency in patient states.

# I LLM USAGE STATEMENT

LLMs were used in this work for polishing the language of the manuscript. The ideas and designs are proposed by the authors.

# J REPRODUCIBILITY STATEMENT

We ensure the reproducibility of our experiments by providing detailed descriptions of our experimental setup, including data preprocessing, model training, and evaluation procedures. All code used in our experiments will be made publicly available upon acceptance of the paper. Datasets derived from MIMIC-IV will be shared in accordance with the data usage agreement of MIMIC-IV and will be accessible via PhysioNet. Datasets can be regenerated by following the data processing scripts provided.

