# OpenReview forum: "Examination Feedback Simulation: Beyond Static and Unique Clinical Trajectories"
_ICLR.cc/2026/Conference — Submitted to ICLR 2026_

### Official Review · Reviewer_pA43 · 2025-10-19

**Soundness:** 2
**Presentation:** 2
**Contribution:** 2
**Rating:** 4
**Confidence:** 3

**Summary:**

This work makes three major contributions in an effort to allow more realistic evaluation of LLMs for sequential clinical decision making, particularly in cases where the desired test by the LLM is not present in the patient's record and as such needs to be generated. It introduces ClinTrack, a new dataset which structures clinical sequences hierarchically and chronologically. It introduces a new metric, SimScore, by which to evaluate this task of novel event generation. Lastly, it introduces ClinSim, a model based on Qwen3-4B which is trained using ClinTrack to generate realistic and appropriate clinical events given a patient's record. With this set-up, the authors report an 11.86% improvement of SimScore using ClinSim to generate new patient events.

**Strengths:**

This work has many strengths. First, it is well motivated that as we evaluate LLMs on real-world tasks, we face the problem that we may incorrectly penalize these agents for not matching real world data although their counterfactual recommendation is valid in its own respect. LLMs are used a lot in healthcare now but we can’t do proper counterfactual analysis when an LLM orders a test without a result, so ClinSim's ability to generate realistic events with an improved performance compared to base models is significant in regards to evaluating models in the future. Further, ClinTrack, ClinSim, and SimScore could each serve as high-significance contributions in their own right, creating a novel dataset, model, and evaluation scoring method for novel data types. Lastly, the authors integration of clinical feedback into the evaluation of their model is a valuable sanity check, allowing users to verify that a clinician deems the output generations of ClinSim to be higher quality than base models, aligning with SimScore judgement.

**Weaknesses:**

The paper suffers from a lack of clarity in the writing/organization as well as a lack of strong baselines for comparison. Generally, the paper would benefit from a running example throughout the text such that the reader can understand what the structure of the data at each point looks like.  For example, Figure 2 is generally confusing and does not clearly explain how ClinTrack is generated. Further, I assume that ClinTrack is used for training and evaluation but even this is not clearly delineated by the authors. Relatedly, all of section 3.2 is difficult to follow without an example of data structure to accompany how ClinSim would actually generate scores in practice. In line with weaknesses of writing/organization, many of the results seem ancillary and either not fully supportive of any main claims and/or not immediately relevant. For example, Figure 4 shows the delta between two Qwen models across different evaluation methods, but the significance of this with respect to SimScore is not fully explained. Similarly for Table 3, if the result of a training ablation is significant to the take-aways of the paper, further discussion/analysis of this pattern is required.

**Questions:**

Is there truly no prior work on simulating missing diagnostic steps or counterfactuals in LLM-based clinical reasoning?

What does “generate samples to be simulated” mean in Figure 2 at step 4?

Why were Levenshtein and ROUGE selected as primary baselines for SimScore?

Why do the authors need SimScore instead of just training a model to generate associated test scores given a test type? It seems that SimScore is a result of the project formulation, but it would simplify the set-up (and thus isolate the experiments) to use a next-event prediction model that is trained to output high-fidelity test scores and then, in deployment, when a non-existent test is ordered it can generate a test result after being trained on what a realistic test result in that setting may be. This seems a simpler approach that does not have as many abstractions-- can you explain why this approach is not sufficient?

Could the authors show a concrete ClinTrack example with a corresponding SimScore computation?

How does the sigmoid function process raw predictions to yield the final similarity score? Generally unsure of how the SimScore is calculated and what the benefits of this recursive approach are as opposed to more general methods of similarity analysis, mean squared error between predicted test result and ground truth test result, or LLM-as-judge?

Given that only one medical expert participated, how reliable is the validation of SimScore alignment?

How does this work connect to broader machine learning interests beyond healthcare (e.g., simulation fidelity, counterfactual reasoning)?

---

> ### Author Response · Authors · 2025-11-22
> **Response to Reviewer pA43 (1/3)**
>
> We sincerely appreciate your helpful comments and valuable questions.
> We hope that the following responses address your concerns.
>
> ## Clarification on Data Structure
>
> ### Example of ClinTrack and SimScore Calculation
>
> > *"Could the authors show a concrete ClinTrack example with a corresponding SimScore computation?"*
>
> Thanks for your valuable suggestion.
> We have added an illustrative example in Appendix D and also included a detailed explanation of the SimScore computation process in Appendix E, which walks through the steps taken to calculate the SimScore.
> ### Further Explanation of Figure 2
>
> > *"What does "generate samples to be simulated" mean in Figure 2 at step 4?"*
>
> Thank you for highlighting this ambiguity.
> This phrase in Step 4 refers to the process of formatting the processed patient records into "History-ExamEvent" pairs to construct our final QA dataset.
>
> To improve clarity, we have revised this text in the figure to read: "**Generate History-ExamEvent Pairs from Trajectories**".
>
>
> ## Clarification on SimScore Design
>
> ### Purpose of SimScore
>
> > *"Why do the authors need SimScore instead of just training a model to generate associated test scores given a test type?"*
>
> Thank you for this insightful question.
>
> First, regarding the **output format**, our intention was not to generate simple numerical "test scores" (which MSE alone could handle). Our goal is to generate detailed diagnostic reports (in structured JSON format). These "reports" are complex and of mixed data-type, containing both:
>
> * **numerical values** (e.g., the value of Hematocrit is 30.1).
> * **textual interpretations** (e.g., the narrative "text" field of a radiology report).
>
> We are focusing on both the precision of 30.1 and the quality of the textual interpretation.
>
> Second, regarding the **task formulation**. We considered alternatives like "next-event prediction" but deliberately adopted a **holistic QA framing** for two key reasons:
>
> 1.  **Leveraging Full LLM Capabilities:** Our QA formulation allows us to leverage the powerful instruction-following, knowledge, and CoT reasoning capabilities of modern, pre-trained QA-oriented LLMs. A raw "next-token prediction" approach (e.g., continuing a JSON string) treats this as a simple regression task on a non-QA distribution, discarding these critical strengths.
>
> 2.  **Maintaining Holistic Integrity:** The other alternative of breaking the report into N tiny QA pairs (e.g., "predict WBC," "predict HGB") would be highly inefficient and would lose the holistic context of the entire report.
>
> Therefore, the JSON format and our SimScore metric are designed precisely to handle this complexity: they unify the evaluation of this single, holistic, structured, and mixed-type output.
>
> ### Baseline Selection for SimScore
>
> > *"Why were Levenshtein and ROUGE selected as primary baselines for SimScore?"*
>
> We selected them as baselines and components for their complementary strengths in evaluating different aspects of our structured text generation task:
>
> 1. **ROUGE** is a standard metric for **Word-level** text generation evaluation. It effectively measures **content overlap** and ensures the presence of key medical terms, relatively independent of their strict order.
>
> 2. **Levenshtein** is a standard metric for **Character-level** text generation evaluation. Complementing ROUGE, it is highly sensitive to **literal form and sequence**.
>
> 3. **Embedding-level metrics (e.g., BERTScore)** were considered but ultimately deemed unsuitable for our specific task for two main reasons:
>    - **Format mismatch**: BERTScore is optimized for natural language prose, whereas our output is **structured JSON text**. Unlike tasks involving unstructured text generation, the high density of symbols (`{`, `}`, `""`, `:`) in our task acts as noise for an embedding-based model.
>    - **Susceptibility to reward hacking & Methodological Unity**: While one could theoretically apply BERTScore only to leaf-node text fields, this is undesirable because we also use our metric as the reward for RL training. Using an LLM-based metric in the RL loop is not only computationally inefficient but also highly susceptible to reward hacking [1].
>
> ---
>
> [1] Wang, Peiyi, et al. "Large language models are not fair evaluators." Proceedings of the 62nd Annual Meeting of the Association for Computational Linguistics (Volume 1: Long Papers). 2024.

---

> ### Author Response · Authors · 2025-11-22
> **Response to Reviewer pA43 (2/3)**
>
> ## Role of the ClinTrack Dataset
>
> > *"I assume that ClinTrack is used for training and evaluation but even this is not clearly delineated by the authors."*
>
> We clarify that "The dataset" in **Section 4.1** refers to the ClinTrack dataset we curated from MIMIC-IV ("We split the dataset into training and evaluation sets with a ratio of 9:1...", Section 4.1), and Figure 3 illustrates this data split.
>
> We have revised the manuscript to make this connection unambiguous, with changes marked in blue for clarity.
>
> ## Significance of Figure 4 (SimScore Sensitivity)
>
> > *"Figure 4 shows the delta between two Qwen models across different evaluation methods, but the significance of this with respect to SimScore is not fully explained."*
>
> Thank you for this question.
> We believe the reference is to **Table 4** (rather than **Figure 4**), as Table 4 "shows the delta between two Qwen models across different evaluation methods".
>
> The significance of this table is to evaluate the **sensitivity** of SimScore against general-purpose metrics like ROUGE and Levenshtein.
>
> Our underlying assumption is that the capability gap between a 1.7B parameter model (Qwen3-1.7B) and a 235B parameter model (Qwen3-235B) should be significant.
> The data in Table 4 shows that while traditional metrics capture this gap, their measured improvement is very small (Δ of +0.0250 to +0.0318).
> In contrast, SimScore shows a much more pronounced improvement (Δ of +0.0757).
>
> This demonstrates that SimScore is more sensitive and more effective at capturing the true magnitude of the quality difference between models.
> The general-purpose metrics, by ignoring the task's structure, appear to underestimate this quality gap.
>
> ## Reliability of SimScore Alignment Validation
> > *"Given that only one medical expert participated, how reliable is the validation of SimScore alignment?"*
>
> Thank you for raising this important concern.
>
> In the revised manuscript, we have expanded our validation study to include two independent raters, and performed a comprehensive analysis of the inter-rater reliability and the alignment between SimScore and the aggregated human consensus, with changes marked in blue for clarity.
> The results, illustrated in Figure 5 and Figure 6, strongly validate the reliability of our metric:
>
> **1. High Human Directional Consistency:**
> Two raters had conflicting judgments on only 3 out of 36 samples (e.g., one preferring Model A while the other preferring Model B).
> * 63.89% of the time, both raters agreed on the exact same outcome (Model A win, Model B win, or Tie).
> * 27.78% of the time, the raters maintained directional agreement, differing only in degree (e.g., one rating a "Win" while the other rating a "Tie").
>
> **2. Strong Alignment with Human Consensus:**
> SimScore demonstrated exceptional alignment with the consensus of the two experts:
> * **High Accuracy:** On samples where experts reached a strong consensus, SimScore correctly identified the winner with 90.48% accuracy.
> * **Discriminative Sensitivity:** SimScore effectively mirrors human confidence. The average score margin for samples with clear human preference was 4 times larger (0.3112 vs. 0.0780) than for ambiguous samples.
>
> **3. Correlation with Consensus Magnitude:**
> To further quantify this, we analyzed the correlation between the Human Consensus Score (aggregated votes ranging from -2 to +2) and the SimScore Difference.
> We found a strong correlation (Pearson $r$ = 0.61, $p < 0.001$; Spearman $\rho$ = 0.64, $p = 0.0016$).
> This demonstrates that SimScore is well-calibrated: it not only aligns with binary human judgments but also scales its score differences proportionally to the degree of human certainty.
>
> ## Prior Work
>
> > *"Is there truly no prior work on simulating missing diagnostic steps or counterfactuals in LLM-based clinical reasoning?"*
>
> Thank you for the question.
>
> We do not claim to be the first to introduce the **notion** of simulating missing diagnostic steps or counterfactuals in LLM-based clinical reasoning, but rather to be the first to formalize it as a distinct **research task** instead of an auxiliary function.
> As we discussed in our related work (Section 2), the prior research SD-Bench [1] has employed language models to simulate feedback for unrecorded examinations.
>
> However, our key contribution is the **systematic formalization of this problem**.
> To the best of our knowledge, our work is the first to systematically define Examination Feedback Simulation as a dedicated research task, providing a clear problem statement, evaluation metrics, a benchmark dataset specifically designed for this purpose, and an exploration of specialized model training strategies.
>
> ---
>
> [1] Nori, Harsha, et al. "Sequential Diagnosis with Language Models." arXiv preprint arXiv:2506.22405 (2025).

---

> ### Author Response · Authors · 2025-11-22
> **Response to Reviewer pA43 (3/3)**
>
> ## Effectiveness of SimScore
>
> > *"...and what the benefits of this recursive approach are as opposed to more general methods of similarity analysis, mean squared error between predicted test result and ground truth test result, or LLM-as-judge?"*
>
> Our data involves mixed data types (numerical values, and free-text interpretations), necessitating a specialized metric that can accurately assess both aspects.
>
> 1.  **General similarity metrics** are designed for general text similarity evaluation, failing to capture the semantic precision of parsing a value and comparing it as a **number** (e.g., distinguishing between "10.5" and "100.5").
>
> 2.  **Mean Squared Error (MSE)** is a powerful metric for simple regression, but it is fundamentally incapable of handling complex free-text components in our diagnostic reports.
>
> 3.  **LLM-as-judge** is computationally expensive and susceptible to reward hacking. In addition, LLMs are known to exhibit strong biases of preferring outputs generated by themselves (or models from their own family) [1-4] and many of our key baselines (e.g., DS-R1-Qwen3-8B, Baichuan-M2, HuatuoGPT-o1) are derived from the Qwen family. Introducing external LLMs as judges (e.g., Gemini or Claude) is ideal, but infeasible due to the privacy requirements of MIMIC-IV.
>
> ---
>
> [1] Laurito, Walter, et al. "AI–AI bias: Large language models favor communications generated by large language models." Proceedings of the National Academy of Sciences 122.31 (2025).
>
> [2] Wataoka, Koki, Tsubasa Takahashi, and Ryokan Ri. "Self-Preference Bias in LLM-as-a-Judge." Neurips Safe Generative AI Workshop 2024.
>
> [3] Panickssery, Arjun, Samuel Bowman, and Shi Feng. "Llm evaluators recognize and favor their own generations." NeurIPS 2024.
>
> [4] Liu, Yiqi, Nafise Sadat Moosavi, and Chenghua Lin. "LLMs as narcissistic evaluators: When ego inflates evaluation scores." ACL 2024.
>
> ## Analysis of Table 3 (Training Ablation)
>
> > *"Similarly for Table 3, if the result of a training ablation is significant to the take-aways of the paper, further discussion/analysis of this pattern is required."*
>
> Thank you for this valuable suggestion.
>
> We have prepared a more unified analysis to replace the fragmented discussion in the revised manuscript, with changes marked in blue for clarity.
> The data in Table 3 reveals how SFT data quality, quantity, and RL methods interact.
> The key takeaways are summarized as follows (Overall SimScore shown in parentheses, Macro SimScore follows the same trend and is omitted here for brevity):
>
> 1.  **Significant SFT Improvement:** Any SFT-based method significantly outperforms the few-shot baseline. This shows that SFT adaptation is a simple but effective way to specialize a pre-trained LLM (52.10%) to our specific task and data format, achieving fairly strong performance (QA: 66.34%).
> 2. **Suboptimal Performance of CoT-only SFT:** CoT training data is generated by existing models, which may suffer from "hindsight bias" (Section 3.3) and "misalignment" of existing model generations (Table 2). As a result, SFT on CoT alone (CoT: 64.73%, FCoT: 61.28%) yields worse performance than SFT on QA data (66.34%).
> 3.  **Quantity vs. Quality of CoT Training Data:** The high-quality but low-volume FCoT data (61.28%) performs worse than the noisy but high-volume CoT data (64.73%). However, FCoT is effective for raising the performance ceiling when used as a supplement (CoT+FCoT: 66.75%).
> 4.  **GRPO as a Finisher:** Finally, GRPO provides a significant boost on top of the best SFT model (from 66.75% to 68.69%), acting as a final "polish" to the model's reasoning abilities.
>
> ## Broader Impact
>
> > *"How does this work connect to broader machine learning interests beyond healthcare (e.g., simulation fidelity, counterfactual reasoning)?"*
>
> Thank you for this insightful question.
>
> Many existing simulation frameworks (e.g., autonomous driving or search), model **deterministic, rule-based physical or logical worlds**, the outcome of a "what if" action is verifiable in principle.
> It can be computed by a physics engine or running a live search.
> A generative simulator is thus an option, often used as an efficient substitute for the real environment, whether the purpose is testing, data augmentation, planning, or reinforcement training.
>
> In contrast, our work models a **non-reproducible complex system**.
> In healthcare, the "what if" ground truth for an unrecorded (counterfactual) action is fundamentally unknowable, as a patient cannot be cloned and re-examined under different conditions to obtain different trajectories.
> A generative simulator is therefore the only way to explore such scenarios.
> This same challenge applies to any domain involving complex socio-economic systems or human-in-the-loop behavioral modeling.
> Our approach, therefore, provides a methodological framework for designing, training, and validating simulators in these critical scenarios where they are a necessity, not merely an option.

---

> > ### Comment · Reviewer_pA43 · 2025-11-27
> >
> > Thank you for clarifying that you aim to show SimScore’s sensitivity to performance differences with Table 4. With this information the table provides a much stronger argument.
> > I still think that stronger baselines are needed beyond Rouge/Levenstein (i.e. model judge) that actually show a model’s raw performance on the prediction task. However, the remaining comments address the majority of my concerns adequately and I adjust my score accordingly.

---

### Official Review · Reviewer_LVTL · 2025-10-27

**Soundness:** 3
**Presentation:** 3
**Contribution:** 3
**Rating:** 6
**Confidence:** 4

**Summary:**

The study tackles a common failure in virtual patient systems: when a clinician orders a reasonable test that wasn’t done in the historical record, the simulator cannot respond. It proposes “Examination Feedback Simulation,” a task that generates plausible results for such tests, and supports it with a curated dataset - ClinTrack, a structure-aware metric - SimScore, and a dedicated model - ClinSim. The scope centers on adult inpatient data from MIMIC-IV and three exam families i.e., radiology, microbiology, labs.

**Strengths:**

- The paper clearly formulates a practical, under-served problem and turns it into a concrete, reproducible task with well-defined inputs and outputs. This clarity makes the contribution easy to evaluate and build upon.
- It contributes an end-to-end pipeline and dataset that convert raw EHR tables into chronological, structured cases, enabling large-scale supervised training for the new task. The json event format is well aligned with clinical exam outputs.
- The SimScore metric evaluates structure, text, and numbers rather than surface text overlap alone. This better matches the nature of medical exam results and encourages faithful formatting.
- A compact, specialized model ClinSim outperforms much larger general models on this task, supported by useful ablations on training strategies. This suggests specialization and metric alignment matter more than sheer scale here.
- The paper provides implementation details, prompts, and an initial blinded clinician preference study improving transparency and offering additional validity for the metric.

**Weaknesses:**

Ordered from the most severe:
- The main evaluation is a proxy, meaning that models are judged against held‑out events that actually happened, while the stated goal is off‑path plausibility. Without large‑scale clinician validation of true counterfactuals, the central claim remains partially untested.
- The same metric - SimScore is used both for optimization (RL) and evaluation, increasing the risk of reward hacking and overfitting to metric quirks. The small human study is not large enough to fully mitigate this concern.
- Ground truth is partially produced via LLM parsing and classification of notes, but there is no reported human audit of extraction accuracy. Label noise could propagate through training and evaluation.
- The project’s scope is limited, the addressed problem is important but narrow, and the data/domain (adult inpatient MIMIC-IV; three exam types) restrict generalizability. It is unclear how well the approach extends to outpatient, pediatrics, or other modalities.

**Questions:**

- How would you evaluate plausibility (clinician‑designed tests that were not performed) to validate the proxy of test on‑path events at scale?
- What safeguards do you use to detect/prevent reward hacking when SimScore is both the training signal and the test metric?
- Since the model labeled parts of the dataset, how accurate are those labels when checked by humans, and how much noise might be in them?
- Why include exact timestamps as inputs for the simulated event and how does performance change if they are removed or perturbed to better show off‑path use?
- Please fix the grammar: "the second best scores are highlighted in underlined".

---

> ### Author Response · Authors · 2025-11-22
> **Response to Reviewer LVTL (1/3)**
>
> We sincerely appreciate your helpful comments and valuable questions.
> We hope that the following responses address your concerns.
>
> ## Q1 & W1: Validation on Counterfactuals
>
> Thank you for this insightful comment.
>
> True validation on counterfactual (off-path) scenarios is indeed a critical challenge for simulation tasks.
> For healthcare scenarios, evaluating "off-path" plausibility is not only a challenge but also almost impossible, as a patient cannot be cloned and re-examined under different conditions to obtain different trajectories.
> The difficulty of obtaining this data is precisely why a robust simulator like ClinSim is needed.
>
> We justify our validation approach with the strict patient-level data split (Section 4.1) and highly consistent parallelism between "on-path" and "off-path" scenarios:
>
> * **In our validation process:**
> A patient is selected from the unseen validation set, an on-path exam order (that was actually performed) is provided along with its preliminary context (masking the result).
> The model receives:
>   * an unseen patient's prior history
>   * a plausible exam request
> * **In the target scenario:**
> A patient is selected, an off-path exam order (that was not actually performed) along with its preliminary context is provided.
> The model receives:
>   * an unseen patient's prior history
>   * a plausible exam request
>
> Therefore, the model is "blind" to the distinction between these scenarios, thus making this proxy the most rigorous quantitative validation possible.
>
> ## Q2 & W2: Reward Hacking in RL Training
>
> Thank you for raising this critical and well-known challenge in reinforcement learning.
> We believe reward hacking has been effectively mitigated in our work through **two primary** safeguards.
> Before detailing these safeguards, we wish to first present **two pieces of empirical evidence** that supports this claim.
>
> ### Q2.1 & W2.1: Empirical Evidence Against Reward Hacking
>
> **1. Ablation Study Evidence (Table 3):**
> * If SimScore were easily "hacked" by exploiting "metric quirks," we would expect the **GRPO-only** training (Row 5) to quickly find these loopholes and achieve a high, "hacked" score.
> The opposite is true: the GRPO-only model (60.69% Overall) performs the **worst** among all methods.
> This strongly suggests SimScore is difficult to optimize and rewards a solid knowledge base (from SFT) rather than superficial "hacks."
> * When GRPO is applied as a final "polishing" step (Row 8 vs. Row 7), the performance improvement is **modest and incremental (+1.94% Overall)**.
> This is characteristic of a genuine fine-tuning process, not the explosive, runaway score inflation one would expect from reward hacking.
>
> **2. Corroboration from External Signals:**
>
> While we agree with the reviewer that our human study (N=36) is "not large enough to **fully** mitigate this concern", its finding of "substantial agreement" is a positive external signal.
> The added inter-rater reliability analysis further strengthens this evidence.
>
> The convergence of our main metric with these independent, external evaluation methods further reduces the likelihood that our results are an artifact of reward hacking.
>
> ### Q2.2 & W2.2: Methodological Safeguards Against Reward Hacking
>
> This successful prevention of reward hacking is attributable to two main factors:
>
> **1. Strict Training and Evaluation Separation (Section 4.1):**
> Since the model is optimized only on training set patients, its high performance on entirely unseen test patients demonstrates a generalizable skill, not an overfit to the metric's quirks.
>
> **2. Robustness of SimScore (Section 3.2):**
> It is a structure-aware, recursive, and composite metric that requires the model to be simultaneously correct on multiple facets, including JSON structure, key matching, text content, and numerical values.
> Exploiting a "quirk" in one component would likely be penalized by another.

---

> ### Author Response · Authors · 2025-11-22
> **Response to Reviewer LVTL (2/3)**
>
> ## Q3 & W3: Potential Noise in Model-labeled Data
>
> Thanks for your valuable question.
>
> To quantify the potential label noise from our LLM-based parsing, we conducted a human audit covering both discharge note extraction and radiology modality classification.
>
> For **discharge note extraction**, we randomly sampled 50 cases from our dataset and manually annotated the ground truth. The LLM-generated labels were compared against these human-annotated labels using ROUGE and Levenshtein similarity. The results show a very high agreement:
>
> | Metric        | ROUGE-1 | ROUGE-2 | ROUGE-L | Levenshtein Similarity |
> | :------------ | :------ | :------ | :------ | :--------------------- |
> | LLM vs. Human | 97.29%  | 90.48%  | 97.29%  | 96.92%                 |
>
> For **radiology modality classification**, we performed a stratified sampling of 125 reports, selecting 25 samples for each of the 5 categories (X-ray, CT, Ultrasound, MRI, and Fluoroscopy).
> Given that radiology reports typically contain explicit modality keywords in their descriptions, this classification task is inherently easy.
> The human audit confirmed an accuracy of **100%** across all sampled cases.
>
> These details have been added to the manuscript (Appendix F). The high performance across both tasks confirms that our parsing pipeline is accurate and valid for training and evaluation.
>
> ## Q4: Clarification and Additional Experiments on Timestamp
>
> Thank you for the question.
>
> ### Q4.1: Clarification on Timestamp Necessity
>
> Precise timestamps are a necessary component of clinical context.
> A patient's state is dynamic; for example, an X-ray showing an acute fracture is clinically distinct from one showing callus formation during healing weeks later.
> Without the timestamp, the model lacks the information needed for fine-grained modulation of the patient's state.
>
> To quantify this temporal dependency, 3,000 patients were randomly sampled.
> We computed the Pearson correlation between the time gap (in hours) and the similarity in exam results (SimScore) for the first and last same-type events within each admission.
>
> | Event Type        | Count | Pearson R | P-Value  | Slope     |
> | :---------------- | :---- | :-------- | :------- | :-------- |
> | LabEvent          | 4,094  | -0.101599 | 7.24e-11 | -4.86e-05 |
> | MicrobiologyEvent | 1,129  | -0.114982 | 1.08e-04 | -2.48e-05 |
> | RadiologyEvent    | 2,305  | -0.178300 | 6.42e-18 | -5.05e-05 |
>
> The results indicate a statistically significant negative correlation across all exam types (p < 0.001).
> The extremely low p-values confirm the certainty of this temporal drift, while the correlation magnitude reflects the gradual nature of patient state evolution.
>
> ### Q4.2: Ablation Study on Timestamp
>
> We conducted an ablation study examining two types of temporal modifications:
>
> * **Timestamp Removed:** The target timestamp is removed from the input.
>
> * **Timestamp Perturbed:** Two specific temporal shifts are performed: (1) backward: setting the target timestamp to be concurrent with the last observed event in the patient's history, and (2) forward: postponing the target timestamp by one day, seven days and thirty days respectively.
>
> The value similarity of event_time is set to 1.0 in this ablation study to ensure a fair comparison.
>
> | Setting                                | Radiology (%) | Microbiology (%) | Lab (%) | Overall (%) | Macro (%) |
> | :------------------------------------- | ------------: | ---------------: | ------: | ----------: | --------: |
> | With Timestamp                         |         64.74 |            95.19 |   68.78 |       68.77 |     76.24 |
> | Timestamp Removed                      |         64.62 |            95.09 |   68.48 |       68.60 |     76.06 |
> | Timestamp Perturbed, backward          |         64.59 |            95.13 |   68.10 |       68.48 |     75.94 |
> | Timestamp Perturbed, forward (1 day)   |         64.39 |            94.87 |   68.18 |       68.35 |     75.82 |
> | Timestamp Perturbed, forward (7 days)  |         64.09 |            94.68 |   67.86 |       68.06 |     75.54 |
> | Timestamp Perturbed, forward (30 days) |         64.12 |            94.59 |   67.62 |       68.00 |     75.45 |
>
> As shown in the table, all modifications (Removing or Perturbing) cause slight drops in performance, with larger perturbations leading to larger deviations.
> This indicates that ClinSim is robust and does utilize timestamp information to some extent for fine-grained simulation, consistent with the gradual evolution observed in Q4.1.
> We have updated the manuscript to include this discussion in Appendix H.

---

> ### Author Response · Authors · 2025-11-22
> **Response to Reviewer LVTL (3/3)**
>
> ## W4: Clarification on Generalizability
>
> Thank you for this comment on the project's scope.
>
> Our work is focused on adult inpatient data from MIMIC-IV and thus inherits its limitations.
> However, this scope was a deliberate choice guided by three main considerations:
>
> 1.  **Data Availability:** Our reliance on MIMIC-IV stems from its status as one of the largest and most widely-used EHR datasets available, which is essential for reproducible research on this new task.
> 2.  **Exam Coverage:** The three exam types (radiology, labs, microbiology) are the only **structured** exams available in MIMIC.
> We acknowledge the necessity of expanding to other exam types in future work, but argue they are highly representative and cover a wide array of common clinical scenarios, thus providing a solid foundation for this initial study.
> Other exam types (e.g., ECG) are provided in discharge notes as unstructured text; therefore future work can be extended to these via careful information extraction.
> 3.  **Task-Setting Fit:** Although we recognize the need to extend to outpatient settings in future work, we argue that for inpatient and outpatient settings, the need for a simulation model differs significantly.
> For outpatient care, which often follows more single-visit, fixed workflows, the extent of "longitudinal" patient history is limited, making the need for a simulator less acute.
>
> Most importantly, our framework was **intentionally designed for extensibility**.
> Our contribution of a JSON-based format and the structure-aware SimScore is inherently generalizable.
> To extend to new databases and new data types (e.g., new modalities like actual radiology images instead of text reports), our framework can easily accommodate this by simply integrating a new data-specific metric (e.g., an image similarity score) as a new "leaf node" in the SimScore's evaluation tree.
> The core methodology remains unchanged.
>
> We have clarified this in the future work (Section 5.2) of our revised manuscript, with changes marked in blue for clarity.
>
> ## Q5: Grammar Fix
>
> We are grateful for your attention to detail.
> We have corrected the sentence to read: "the second-best scores are underlined" in the revised manuscript, marked in blue for clarity.

---

### Official Review · Reviewer_SRPV · 2025-10-29

**Soundness:** 2
**Presentation:** 3
**Contribution:** 2
**Rating:** 4
**Confidence:** 4

**Summary:**

This paper addresses a limitation in current LLM-based clinical simulations: their reliance on static EHRs, which causes simulations to fail when an LLM orders a medically valid but unrecorded examination. The study formulates a new task "Examination Feedback Simulation" to dynamically generate plausible results for such off-path orders. To support this, the study introduces three main contributions: (1) ClinTrack, a dataset curated from MIMIC-IV; (2) SimScore, an evaluation metric to assess the quality of the simulated JSON-based examination results; and (3) ClinSim, a generative model to perform the task. The paper demonstrates that ClinSim significantly outperforms baseline models on multiple tasks, providing a foundation for dynamic virtual patient simulations.

**Strengths:**

1. The paper addresses multiple facets of the challenge. It introduces a new benchmark dataset (ClinTrack), a novel evaluation metric (SimScore), and a specialized model (ClinSim) to perform the defined task. The connection between these three components is logical and well-executed.
2. The methodology is clear. The process for building the dataset and the SimScore metric is clear and easy to follow.
3. The proposed ClinSim model shows significant performance improvement over existing baseline LLMs. The inclusion of a wide range of baseline models in Table 1 provides a strong empirical validation.

**Weaknesses:**

1. I find the distinction between the “static” and “dynamic” aspects of the work somewhat unclear. The proposed task involves dynamic simulation of new feedback, yet the underlying dataset (ClinTrack) is inherently static, representing historical records. It would be helpful for the authors to clarify how this dynamic component is operationalized. Specifically, how much historical data is required to generate reliable simulation results, and how many simulated trajectories are produced per patient record?
2. I'm wondering if the proposed SimScore fully captures clinical utility. For example, is a simulation with a high SimScore guaranteed to lead to a better diagnosis? The alignment with expert judgment (89%) is very promising, but I'd be interested in the authors' thoughts on how fidelity (which SimScore measures) relates to downstream clinical settings or diagnostic accuracy.
3. The paper introduces a model called ClinSim, described as being built upon Qwen3, using Qwen3-4B for training and Qwen3-30B for parsing clinical notes. From the description, it seems that ClinSim primarily involves fine-tuning or adapting existing Qwen models. I am not very convinced if this is a significant contribution of model development.

**Questions:**

1. The paper notes that the authors treat the output as plain text, potentially overlooking the intrinsic structure and semantics of the JSON data. Could the authors elaborate on why preserving the JSON format might be important or beneficial in this context?
2. For certain applications, such as general simulations, generating synthetic outputs may be acceptable. However, for more sensitive areas like disease diagnosis, could such simulations pose risks or lead to misleading conclusions?

**Details Of Ethics Concerns:**

I don't see any ethical concerns in this paper, but it could be helpful to clarify the data access policy.

---

> ### Author Response · Authors · 2025-11-22
> **Response to Reviewer SRPV (1/2)**
>
> We sincerely appreciate your helpful comments and valuable questions.
> We hope that the following responses address your concerns.
>
> ## W1: Clarification on "Dynamic Simulation" vs. "Static Dataset"
>
> Thank you for this insightful question. The distinction lies in the separation between the **static training resources** and the **dynamic inference capabilities** of the ClinSim model.
>
> **1. Static Dataset:** ClinTrack dataset is indeed inherently static, representing historical ground truth. In our implementation, we input the entire available history (up to a 4,096-token context window) as the clinical context to exploit all relevant information for reliable simulation.
>
> **2. Learning from Static Data:** We maximize the utility of the static records by decomposing them into discrete "(History, ExamOrder) $\to$ ExamFeedback" training samples. This allows the model to learn the latent medical distribution and acquire the capability to generate plausible exam feedback for a given context.
>
> **3. Dynamic Model:** The "dynamic" aspect emerges during the model's deployment as a component of a virtual patient. When a doctor model orders an examination that diverges from the original static trajectory, ClinSim dynamically generates a plausible result **in real-time** to bridge the gap. This ensures the interaction continues fluently without being halted by the limitations of the static record, rather than pre-generating multiple synthetic trajectories per patient.
>
> ## W2: Clarification of the Clinical Utility of SimScore
>
> Thank you for your question.
>
> SimScore is designed to measure the fidelity (i.e., the plausibility and realism) of the simulated results, not the downstream clinical utility (e.g., diagnostic accuracy).
> A high SimScore does **not** inherently imply a better diagnosis.
>
> Our central argument is that high fidelity is a necessary prerequisite for any meaningful evaluation of downstream clinical utility.
> It is impossible to fairly evaluate an LLM doctor's diagnostic skill using a simulation that provides low-fidelity, nonsensical, or "garbage-in" feedback.
>
> Therefore, the primary goal of this foundational work was to first solve this problem: how to generate, and reliably measure, high-fidelity feedback.
>
> Evaluating the downstream diagnostic accuracy using our high-fidelity simulator is precisely the logical next step that our work enables.
> This aligns perfectly with our stated Future Work (Section 5.2) for evaluating clinical decision-making utility.
>
> ## W3: Significance of Model Development Contribution
>
> Thank you for your valuable comment.
>
> Our contribution to model development lies in conducting extensive experiments for a better development path for training ClinSim.
>
> Our experiments reveal a critical insight:
> Standard model development assumes that CoT reasoning enhances performance on complex tasks almost unconditionally.
> However, Table 2 and Table 3 reveal that direct application of CoT can be suboptimal for this task (e.g. due to "misalignment"), where localized clinical logic is required rather than general reasoning patterns.
>
> The exploration not only guides our model development but also provides critical experimental data and theoretical analysis.
>
> ## Q1: The Importance of JSON Format
>
> Thank you for the question.
>
> The JSON format is necessary to represent the complexity of our target output, which is inherently structured and contains mixed data types:
> * **numerical values** (e.g., the value of Hematocrit is 30.1).
> * **textual interpretations** (e.g., the narrative "text" field of a radiology report).
>
> JSON allows us to separate and evaluate these components appropriately, while plain-text metrics (like ROUGE or Levenshtein) perform well on the textual parts but fail to capture the semantics of numerical values.
>
> When comparing two numerical results, such as 100.5 and 10.5, medical applications require understanding their numerical similarity, which is $1-\frac{|100.5-10.5|}{\max(|100.5|,|10.5|)}=10.45$%.
> In contrast, ROUGE-1 (50%), ROUGE-2 (0%), ROUGE-L (50%), and Levenshtein similarity (88.9%) are all uninformative in this context.
>
> The JSON format allows us to apply the correct, medically-meaningful numerical comparison easily.

---

> ### Author Response · Authors · 2025-11-22
> **Response to Reviewer SRPV (2/2)**
>
> ## Q2: Safety Implications and Risk Mitigation
>
> Thank you for this crucial question regarding safety.
>
> 1.  **Indirect Application (Safety Sandbox):** ClinSim is designed as an offline evaluation virtual patient for assessing medical LLMs under test, not as an online clinical decision support system deployed directly to patients. Any "risk" is fully controlled by the evaluator and remains strictly confined within the evaluation sandbox.
>
> 2.  **Prevalence in Medical Research:** Leveraging synthetic data is a widely accepted practice in medical AI. As evidenced by recent literature, synthetic EHR data is extensively used for training predictive models, data augmentation, and algorithmic validation [1-5]. Our work extends this established paradigm from "data augmentation" to "interactive evaluation."
>
> 3.  **The Greater Risk of Static Evaluation:** In fact, our primary motivation is to step towards safety, rather than away from it. Relying solely on static EHRs for evaluation poses a higher level of risk. Models evaluated by such approaches may achieve high scores by overfitting to specific recorded trajectories without possessing true clinical reasoning capabilities [5]. Thus, we view the simulator as a necessary mechanism to mitigate deployment risks.
>
> ---
>
> [1] Theodorou, Brandon, Cao Xiao, and Jimeng Sun. "Synthesize high-dimensional longitudinal electronic health records via hierarchical autoregressive language model." Nature communications 14.1 (2023): 5305.
>
> [2] Loni, Mohammad, et al. "A review on generative AI models for synthetic medical text, time series, and longitudinal data." npj Digital Medicine 8.1 (2025): 281.
>
> [3] Ibrahim, Mahmoud, et al. "Generative AI for synthetic data across multiple medical modalities: A systematic review of recent developments and challenges." Computers in biology and medicine 189 (2025): 109834.
>
> [4] Chen, Anjun, and Drake O. Chen. "Simulation of a machine learning enabled learning health system for risk prediction using synthetic patient data." Scientific Reports 12.1 (2022): 17917.
>
> [5] Chen R J, Lu M Y, Chen T Y, et al. Synthetic data in machine learning for medicine and healthcare[J]. Nature Biomedical Engineering, 2021, 5(6): 493-497.
>
> ## Response to Ethics Concerns
>
> Thank you for raising the important ethical consideration.
>
> The lowest barrier to entry for the ClinTrack dataset is having a valid MIMIC-IV data use agreement, without requiring any separate or additional data access permissions.
> This is because all data curation and processing scripts (e.g., pipeline/psql2patients.py, pipeline/disch_parse.py, pipeline/radio_parse.py) have been included in the supplementary materials submitted with our paper, and are already available.
> These scripts simply require minimal configuration (e.g., local database parameters) and for the user to populate the few-shot example placeholders from their own local MIMIC-IV data.
>
> Furthermore, as we have detailed in Appendix F, we have conducted experiments to validate the LLM-based parsing steps and confirmed that this process has a very high degree of reproducibility.

---

### Official Review · Reviewer_12P8 · 2025-11-02

**Soundness:** 2
**Presentation:** 2
**Contribution:** 2
**Rating:** 2
**Confidence:** 4

**Summary:**

This paper introduces Examination Feedback Simulation, a new task for generating plausible exam results to overcome the static nature of clinical trajectories in virtual patients. The authors build ClinTrack, a hierarchical, time-ordered dataset from MIMIC-IV, and propose SimScore for evaluating simulation quality. Their model, ClinSim (4B parameters), outperforms LLMs up to 235B parameters by over 10 points in SimScore, enabling more dynamic and realistic healthcare simulations.

**Strengths:**

1. The newly curated ClinTrack dataset and benchmark is valuable for the research of clinical trajectory simulation.
2. A new rule-based metric is proposed considering expert judgement in human evaluation.

**Weaknesses:**

1. The novelty of this work seem limited. The major contributions are the dataset and benchmarks. Please further clarify the motivation and novelty.
2. In introduction, the third contribution is overclaimed. There seems no change in architecture. The model is only pretrained on more specialized data. This is unfair to directly compare it with other existing models.
3. It is unclear why the proposed ClinSim, which is based on Qwen3, outperform Qwen3 by a large gap. It seems only the curated data is contributing.
4. There is no reference or detailed introduction of the database MIMIC-IV, which confuses readers.
5. No new training paradigm is proposed specialized for the EXAMINATION FEEDBACK SIMULATION in this work. The author is only validating the contribution of existing paradigms, such as SFT, CoT, GRPO.
6. The presentation of Figure 3 is poor, such as the organization of trainset and valset, the lack of detailed description of different event.
7. There is no discussion on how to solve hallucinations in clinical trajectory simulation.

**Questions:**

Please see the weaknesses

**Details Of Ethics Concerns:**

Discrimination / bias / fairness concerns in the curated ClinTrack datasets.

For example
Demographic Representation Bias: The training EHR data may contain far fewer elderly or minority patients. As a result, simulated lab values or imaging findings for these groups could be unrealistic or systematically inaccurate.

Historical Treatment Bias: If real-world clinicians historically under-ordered certain tests for women or uninsured patients, the model may learn to simulate fewer or different examinations for these groups — reinforcing past inequities.

Unequal Simulation Quality: The generated “feedback” (e.g., lab test results) might have higher error rates or hallucinations for specific demographic groups, leading to unfairly poorer virtual care trajectories.

---

> ### Author Response · Authors · 2025-11-22
> **Response to Reviewer 12P8 (1/2)**
>
> We sincerely appreciate your helpful comments and valuable questions.
> We hope that the following responses address your concerns.
>
> ## W1: Task Formulation Beyond Data and Benchmark
>
> Thank you for the comment.
>
> We clarify that our motivation for this work is to address the problem that "existing simulation frameworks inherit the limitation of clinical trajectory uniqueness naturally present in static EHRs" (Section 1, Paragraph 2).
>
> Our primary novelty is "to make the first attempt to systematically define the **task** of examination feedback simulation" (Section 1, Paragraph 4) rather than constructing a dataset alone.
> We elevate it to a distinct research task with clear definitions and input-output specifications.
>
> Our careful data construction, tailored evaluation metric, and specialized model are all in service of this new task.
>
> ## W2 & W3 & W5: Training Strategy Towards Stronger Performance
>
> Thank you for your comments.
>
> First, we wish to clarify that ClinSim is optimized via **"supervised fine-tuning"** and **"reinforcement learning"** (Section 4.1), rather than being **"pretrained"** as mentioned in the review.
>
> Second, our experiments demonstrate that simply applying existing methods or adding specialized data does not necessarily lead to better performance.
> Table 3 shows that simply using CoT data (64.73%) or Filtered CoT alone (61.28%) performs even worse than non-CoT pairs (66.34%).
> It indicates that relying solely on the curated data is inadequate.
> Our contribution lies in identifying that a specific path (CoT $\rightarrow$ Filtered CoT $\rightarrow$ GRPO) is required to minimize the impact of suboptimal samples in the crafted CoT data (Section 4.2), retain reasoning logic and exposure to diverse cases, and gradually refine the model towards precise clinical logic.
>
> ## W4 & W6: MIMIC-IV Context and Visualization
>
> Thank you for your comments.
>
> We introduce the database in the first paragraph of Section 3.1, but we agree that a formal reference is needed.
> We have added a formal citation for MIMIC-IV in the revised manuscript, with changes marked in blue for clarity.
>
> To better illustrate the data splits in Figure 3, we have added an illustrative example of the data structure in Appendix D for clarity.

---

> ### Author Response · Authors · 2025-11-22
> **Response to Reviewer 12P8 (2/2)**
>
> ## W7: Hallucination
>
> Thank you for your comment.
>
> We clarify that addressing hallucination is intrinsic to our methodology, utilized through our SimScore-guided training framework.
>
> In the context of examination feedback, "hallucination" manifests as factual inconsistency or clinical implausibility in the simulated results.
> Our solution is embedded directly in our evaluation and training framework:
>
> * **Evaluation Necessity:** The SimScore metric is explicitly designed to measure factual fidelity. Standard metrics (ROUGE/Levenshtein) are insufficient as they treat outputs as pure text and fail to account for the mixed data types (numerical and textual) and critical structural elements required for clinical validity.
>
> * **Optimization Alignment:** Our training strategy overcomes the inherent limitation of SFT: SFT minimizes token-by-token prediction error (a local optimum), but this cannot guarantee that the resulting sequence satisfies the global structural and factual constraints of the examination report.
> By utilizing SimScore-guided Reinforcement Learning, we shift the optimization objective from general prediction (including CoT) to utility maximization based on the final, globally verified output.
> This enforces factual consistency and minimizes hallucination.
>
> ## Ethics: Distinguishing Simulation Fidelity from Decision Bias
>
> Thank you for the comment.
>
> We must first clarify that the expected function of our model is solely to generate the result ($\hat{E}_i$) given a specific examination order ($ \text{Type}(E_i) $ and $ \text{Time}(E_i) $) from the LLM Doctor being evaluated.
> ClinSim is a patient simulator and the decision to **order** an exam is **outside its scope**.
> The decision to order tests remains the exclusive function of the LLM Doctor.
>
> * **Evidence of Equal Simulation Quality:**
> We conducted a dedicated Demographic Subgroup Analysis in the validation set, and the results below demonstrate that ClinSim’s SimScore performance is generally consistent across demographics, showing no significant bias against specific groups.
> We have also added this analysis to Appendix G in the revised manuscript.
>
> * **Transparency and Auditability:**
> The retention of the model’s Chain-of-Thought reasoning provides an essential audit mechanism.
> This transparency illuminates the model’s simulation logic, helping to identify and verify whether hidden biases or flawed reasoning paths influenced the generated clinical outcome.
>
> | Race     | Count  | Overall | Overall STD | Macro  | Macro STD |
> | :------- | :----- | :------ | :---------- | :----- | :-------- |
> | Asian    | 958    | 0.7014  | 0.1836      | 0.7713 | 0.1787    |
> | Black    | 2,314  | 0.6875  | 0.1731      | 0.7640 | 0.1678    |
> | Hispanic | 1,009  | 0.6920  | 0.1834      | 0.7606 | 0.1664    |
> | White    | 15,663 | 0.6852  | 0.1720      | 0.7616 | 0.1656    |
> | Unknown  | 2,106  | 0.6897  | 0.1855      | 0.7544 | 0.1681    |
>
> | Age Group           | Count | Overall | Overall STD | Macro  | Macro STD |
> | :------------------ | :---- | :------ | :---------- | :----- | :-------- |
> | 18-39 (Young Adult) | 3,191 | 0.6986  | 0.1829      | 0.7714 | 0.1653    |
> | 40-64 (Middle Age)  | 8,437 | 0.6960  | 0.1833      | 0.7641 | 0.1686    |
> | 65-79 (Senior)      | 5,857 | 0.6827  | 0.1706      | 0.7564 | 0.1586    |
> | 80+ (Geriatric)     | 4,565 | 0.6673  | 0.1538      | 0.7548 | 0.1745    |
>
> | Gender | Count  | Overall | Overall STD | Macro  | Macro STD |
> | :----- | :----- | :------ | :---------- | :----- | :-------- |
> | M      | 11,104 | 0.6878  | 0.1761      | 0.7601 | 0.1672    |
> | F      | 10,946 | 0.6860  | 0.1728      | 0.7630 | 0.1664    |

---

### Author Response · Authors · 2025-12-01
**General Responses and Summary of Revisions**

We sincerely thank all reviewers for their feedback, which has helped us significantly improve our work.
We are encouraged by the reviewers' recognition of several key strengths:

- "*The newly curated ClinTrack dataset and benchmark is valuable for the research of clinical trajectory simulation.*" (Reviewer 12P8)
- "*It introduces a new benchmark dataset (ClinTrack), a novel evaluation metric (SimScore), and a specialized model (ClinSim) to perform the defined task. The connection between these three components is logical and well-executed.*" (Reviewer SRPV)
- "*The paper clearly formulates a practical, under-served problem and turns it into a concrete, reproducible task with well-defined inputs and outputs. This clarity makes the contribution easy to evaluate and build upon.*" (Reviewer LVTL)
- "*First, it is well motivated...ClinTrack, ClinSim, and SimScore could each serve as high-significance contributions in their own right, creating a novel dataset, model, and evaluation scoring method for novel data types.*" (Reviewer pA43)

Based on their constructive comments, we have made substantial improvements to strengthen our paper's contributions and address the limitations. The major enhancements include:

- **Extended Human Evaluation.** We have conducted an expanded expert study confirming SimScore's alignment with human consensus (Section 4.3), and a human audit of the parsing pipeline confirming the integrity of the ClinTrack dataset (Appendix F).
- **Additional Experiment Results.** We have added a timestamp ablation study demonstrating the model's temporal sensitivity (Appendix H), and a demographic subgroup analysis verifying the model's robustness and fairness across race, age, and gender (Appendix G).
- **Refined Training Dynamics Discussion.** We have unified the analysis of SFT, CoT, and GRPO to provide a clearer insight into how data quality and reinforcement learning interact to achieve superior performance (Section 4.2).
- **Enhanced Data Documentation & Examples:** We have significantly improved clarity by providing a data structure example and a step-by-step breakdown of the SimScore computation (Appendix D and E).

We thank the reviewers again for their valuable insights and suggestions that have helped us improve the work.
We believe these additions have further realized the potential of our framework and established a solid foundation for examination feedback simulation, paving the way for future research in dynamic clinical evaluation.

---

### Meta-Review · Area_Chair_Ftgx · 2026-01-06

**Summary:**

The reviewers' concerns mainly lie in the contributions and some unverified claims, as well as the usability and generalizability of the proposed method in practice. The authors provided further elaboration in the rebuttal, but as stated by reviewer pA43, this paper needs more effective comparisons and more detailed experiments. Thus, I recommend that the paper should be revised and resubmitted with more comprehensive evaluation.

**Reviewer Concerns:**

Concerns raised by Reviewer 12P8 and Reviewer pA43 are still outstanding, and I concur that the paper requires futher extensive and rigorous evaluation.

**Reviewer Scores:**

The reviewers may maintain their scores, especially Reviewer 12P8.

---

### Decision · Program_Chairs · 2026-01-26

Reject